# Large mechanical properties enhancement in ceramics through vacancy-mediated unit cell disturbance

Zhuo Chen[1], Yong Huang[1], Nikola Koutná [2,3], Zecui Gao[2],
Davide G. Sangiovanni [3], Simon Fellner [1], Georg Haberfehlner [4],
Shengli Jin [5], Paul H. Mayrhofer [2], Gerald Kothleitner [4,6] &
Zaoli Zhang [1,7]✉

Tailoring vacancies is a feasible way to improve the mechanical properties of ceramics. However, high concentrations of vacancies usually compromise the strength (or hardness). We show that a high elasticity and flexural strength could be achieved simultaneously using a nitride superlattice architecture with disordered anion vacancies up to 50%. Enhanced mechanical properties primarily result from a distinctive deformation mechanism in superlattice ceramics, i.e., unit-cell disturbances. Such a disturbance substantially relieves local high-stress concentration, thus enhancing deformability. No dislocation activity involved also rationalizes its high strength. The work renders a unique understanding of the deformation and strengthening/toughening mechanism in nitride ceramics.

Defects in crystalline materials are atomic compositions or arrangements that deviate from the periodic structure of an ideal crystal, encompassing zero to three-dimensional defects. It is widely believed that one-dimensional defects (dislocations) are the main carriers of crystal plastic deformation[1,2], relaxing long-range strain through nucleation and movement. In ceramic materials, however, strong covalent bonds cause extremely high critical resolved shear stress (CRSS)[3], together with few slip systems that operate at room temperature, which greatly limits their deformation via dislocation movement. Consequently, significant stress concentration events make ceramics very brittle and fragile. Can the stress concentration be alleviated by other defects? If feasible, this could primarily improve the deformability and toughness of ceramics. One potential solution is the design of porous structures[4–6]. When subjected to external stress, pores can absorb deformation energy[7,8] through mechanisms such as shrinkage or rearrangement. This behavior helps alleviate the impact of stress concentrations and crack propagation, thereby enhancing the

materials' mechanical properties. Unfortunately, brittle ceramic materials make it difficult to undergo plastic deformation around voids. When voids serve as stress concentration zones and sites for crack initiation[9], they can compromise the overall mechanical properties.

Further reduction of the void sizes down to atomic-scale vacancies has more potential for enhancing ceramic toughness and deformation ability. Recent experimental and theoretical studies have shown that ceramic materials with zero-dimensional defects (vacancy) can significantly improve their toughness and may generate potential plasticity[10–14]. However, current studies have only attributed the toughening effect to enhancing intrinsic fracture properties or through vacancy-induced phase transformation toughening[10–14]. Here, we propose that a high density of disordered vacancies is a solution to improve the deformability of ceramics, similar to a unit-cell scale "porous structure", in which introducing N vacancies can enhance deformation ability through unit-cell distortion or atomic rearrangement near the vacancies.

[1]Erich Schmid Institute of Materials Science, Austrian Academy of Sciences, A-8700 Leoben, Austria. [2]Institute of Materials Science and Technology, TU Wien, A-1060 Vienna, Austria. [3]Department of Physics, Chemistry, and Biology (IFM), Linköping University, Linköping SE-58183, Sweden. [4]Institute of Electron Microscopy and Nanoanalysis, Graz University of Technology, Steyrergasse 17, 8010 Graz, Austria. [5]Chair of Ceramics, Montanuniversität Leoben, Peter-Tunner Strasse 5, 8700 Leoben, Austria. [6]Graz Centre for Electron Microscopy, Steyrergasse 17, 8010 Graz, Austria. [7]Department of Materials Science, Montanuniversität Leoben, Franz-Josef-Strasse 18, 8700 Leoben, Austria. ✉e-mail: zaoli.zhang@oeaw.ac.at

In this work, by exploiting the superlattice (SL) architecture, we stabilized the metastable cubic (fcc) $WN_x$ with high concentration (up to ~50%) of disordered anion vacancies (abbr. "vac") through the template effect from highly structurally stable and perfectly lattice-matched fcc phase of TiN (producing essentially zero lattice mismatch, hence minimizing interface effects). Due to the significant vacancy content in $WN_x$ ($x = 0.5$), the SL layers exhibited severe local distortions. Microcantilever experiments confirmed that such ceramic's high strength (~9 GPa) and bending strain (~2.8%). Employing spherical aberration (Cs)-corrected transmission electron microscopy (TEM) and atomic-scale simulations, we demonstrate that the mechanical response of $WN_x$ is dominated by a unit-cell disturbance (which implies that further unit-cell distortions occur during deformation), which acts as a deformation carrier and simultaneously prevents high stress concentrations. The enhanced mechanical properties of $WN_x$ with ultra-high concentration (~50%) of disordered vacancies can be attributed to the effects of the unit-cell disturbance absorbing deformation energy, presence of N vacancies increasing the valence electron concentration per unit cell, and unit-cell distortion hindering dislocation nucleation.

## Results

### Unique microstructure

Through atomic-resolution TEM observations and DFT (density functional theory) simulations, we studied the unique microstructure of $WN_x$ in detail. HAADF image (high-angle annular dark-field, Fig. 1a) depicts the monocrystalline SL with a bilayer thickness of ~10 nm (Supplementary Fig. 1), with the bright contrast layer being $WN_x$. Figure 1b shows three core-level energy-loss spectra recorded from the $WN_x$ and TiN layers. In $WN_x$, the core-loss spectra indicate that the second peak of the corresponding N-K edge broadens and decreases in intensity, ascribing to the N-vacancy-induced effect[15]. The relative peak shift also consolidates a high concentration of N-vacancy in $WN_x$ layers. Additional evidence of different vacancy contents in WN and TiN is shown by elemental chemistry analysis by EDAX (energy-dispersive X-ray spectroscopy, Supplementary Fig. 2).

We imaged N-vacancies on the atomic scale using Cs-corrected high-resolution TEM (HRTEM). The high-resolution atomic image (Fig. 1c) along the <100> direction (where W and N atom columns overlap) shows that $WN_x$ has a big atom column intensity difference and additional contrast between the W/N atom columns, which is significantly different from the adjacent TiN layer. Comparing experimental results with image simulations (Fig. 1c, Supplementary Fig. 3) confirms that such features are caused by the high vacancy concentration (50% N-vac) in $WN_x$. The disordered N vacancies lead to irregular variations in the intensity of (W/N) atom columns and a change in the local electron scattering conditions that additional contrast appears (Supplementary Fig. 3).

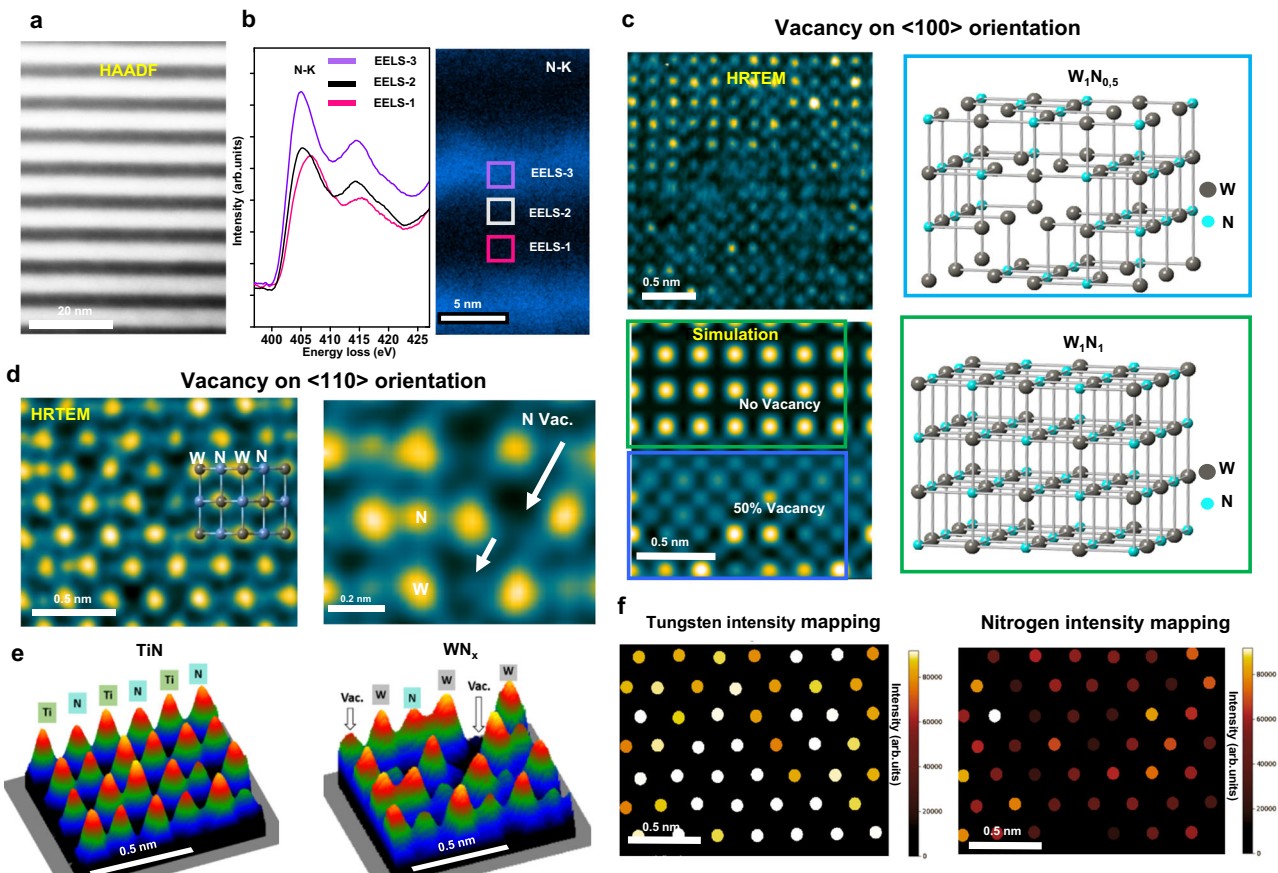

**Fig. 1 | TEM characterization of nitrogen vacancies. a** High-angle annular Dark-Field (HAADF) imaging of $WN_x$/TiN SL (superlattice). **b** EELS (Electron energy loss spectroscopy) results (N-K edge, spectrum, and mapping) of $WN_x$/TiN SL. **c** Detailed experimental observation and image simulation of N vacancy in <100> projection direction. The upper part of the simulated image is a WN without vacancies, and the lower part is a $WN_x$ with 50% disordered vacancies. **d** HRTEM (High-resolution transmission electron microscopy) observation and image simulation of $WN_x$ layer in <110> direction. Note that the schematically indicated positions (arrows) represent a higher concentration of vacancies. **e** Projection images of atom column intensities of TiN and $WN_x$ (HRTEM image display in surface plot mode), respectively. **f** The Gaussian fitting result of Fig. 1d ($WN_x$ layer) shows the signal intensity distribution at the W and N positions, respectively.

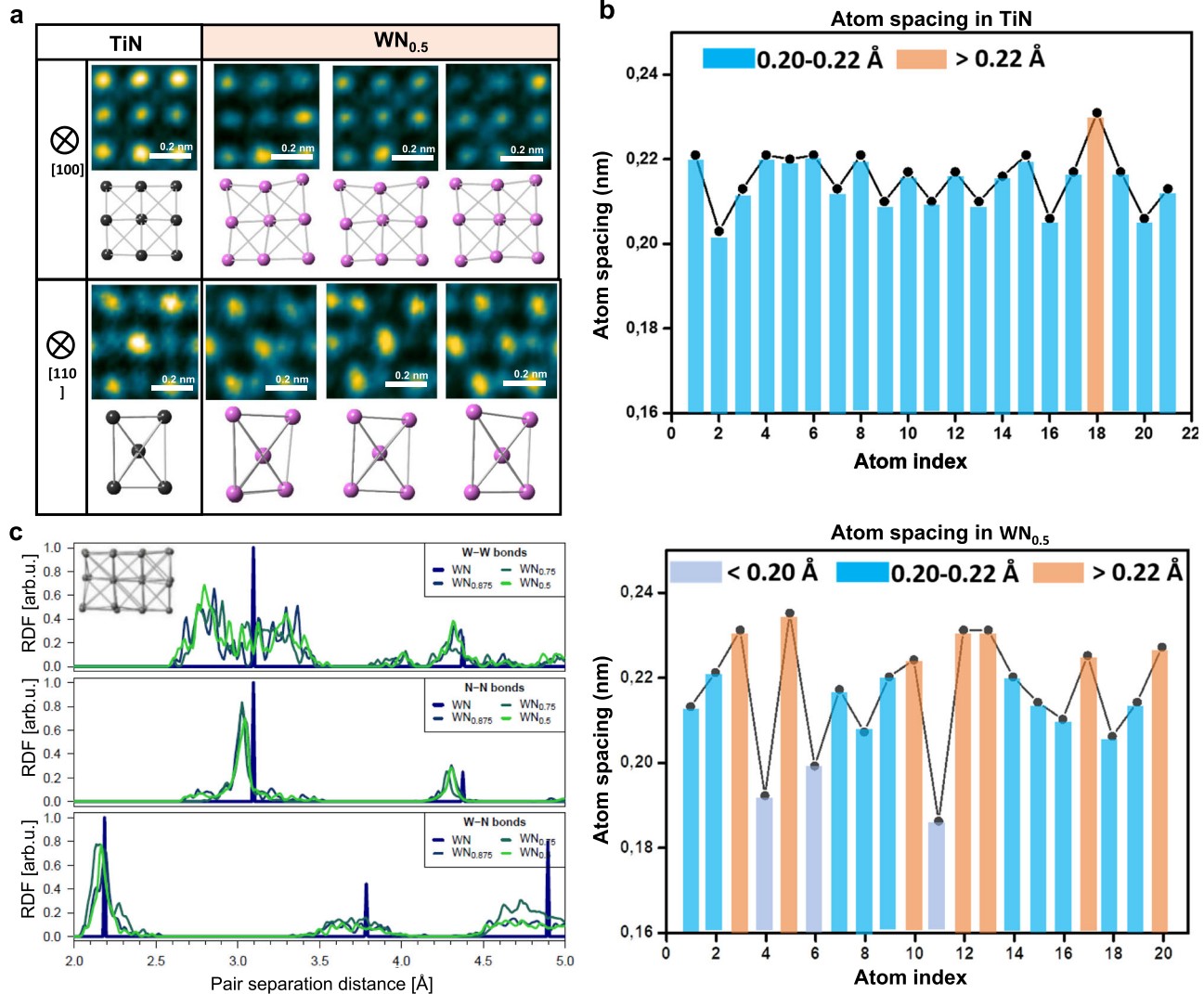

**Fig. 2 | Severe unit-cell distortions in $WN_x$. a** HRTEM observation of the unit-cell distortions of $WN_x$ in the <100> and <110> projection directions, respectively. Only metal atoms are shown in models (beneath the corresponding images). **b** The line profile results (atom spacings) of the $WN_x$ and TiN layer, respectively. The line profiles are obtained from <100> projected HRTEM images and measured along the [010] direction on the (100) plane. **c** Radial distribution functions of W-W, N-N, and W-N bonds in $WN_x$ calculated by DFT (Density-functional theory) simulation. The inserted image shows the atomic model of the DFT simulation of $WN_x$ (<100> projection direction), where only the W atom positions are shown.

N-vacancy can be directly imaged in the <110> projection owing to the separated W and N atom columns. In Fig. 1d ($WN_x$ layer), a significant intensity difference among those N atomic columns can be visible, and those weaker N signal intensities stem from disordered N-vac (Supplementary Fig. 4). A close comparison of the intensity plots shows a more pronounced N-vac present in $WN_x$ than in TiN (Fig. 1e). Gaussian fitting results in $WN_x$ demonstrate that the signal intensity at W positions is uniform (Fig. 1f). In contrast, most of the N column signal intensities is just about half of the locally recorded maximum intensity. These detailed quantitative statistic results (Supplementary Fig. 5) show that the average signal intensity of N columns in the $WN_x$ layer is 48–52% of the maximum N signals, corresponding to 52–48% of N vacancy. Thus, HRTEM analyses are consistent with density functional theory simulations[16], which predicted that $WN_x$ with 50% disordered N-vac can be thermodynamically stabilized in $WN_x$/TiN SL compared to other vacancy configurations.

Although under-stoichiometric $WN_x$ ($x = 0.5$) layers are macroscopically with fcc-structure, they exhibit significant local distortions, as illustrated by $WN_x$ viewed along the [100] and [110] directions (Fig. 2a), where N atom positions clearly deviate from their equilibrium

sites. Contrarily, TiN maintains nearly perfect fcc unit cell. Severe distortion of the W sub-lattice in $WN_x$ is more visible in the [110] projected image (Fig. 2a). Such distortions are also reflected in notable changes of atomic spacings. Severe distortion of the metal sub-lattice in $WN_x$ is also reflected by a large variation of atomic column spacings. $WN_x$ exhibits a standard deviation in atom spacing up to 0.133 Å (Fig. 2b), which is almost twice as large as that in the TiN layer (i.e., 0.066 Å, Fig. 2b).

DFT simulations also point towards significant fluctuations in the metallic atom distances in $WN_x$ (Supplementary Fig. 6). Here, we evaluate and display the lattice distortion via radial distribution functions (RDF). Figure 2c presents a distribution of the first through second nearest neighbor atomic distances of W-W, W-N, and the first through third nearest neighbor atomic distances of N-N, respectively (Supplementary Fig. 7, a simulated image using the calculated model presents a similar distortion to experiments). However, N-N bond distribution demonstrates no significant broadening. Previously, the stoichiometric WN was shown to exhibit both phonon and elastic instabilities[16]. These can be (partially) eliminated via local relaxations towards more stable lower-symmetry phases (e.g., tetragonal,

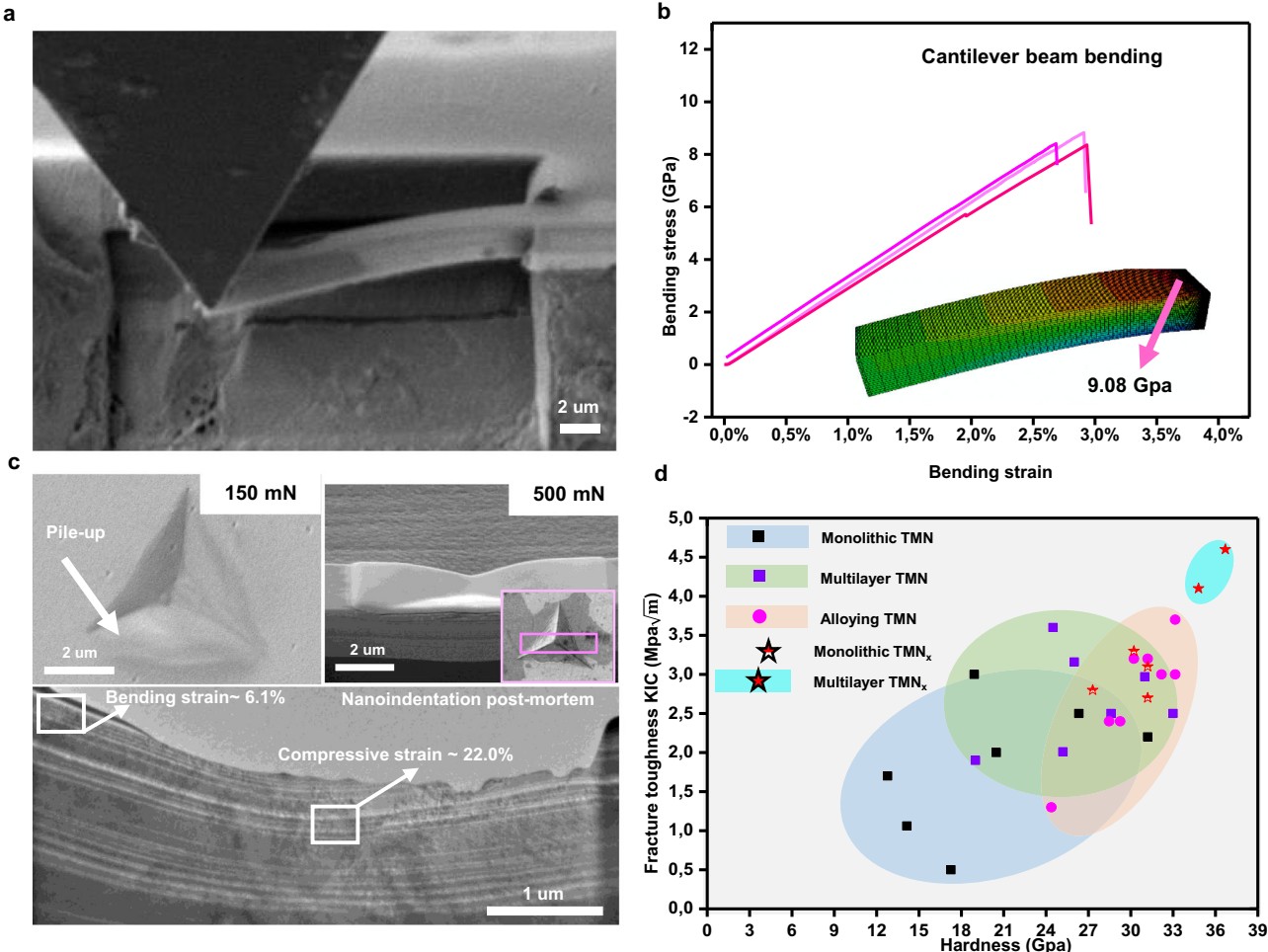

**Fig. 3 | Mechanical properties of WN$_x$/TiN superlattices. a** Cantilever beam bending testing of WN$_x$/TiN superlattices. SEM image shows that SL (superlattice) with a loading depth of 2 μm. **b** Stress-strain curves for WN$_x$/TiN cantilever bending (three colors from three samples). Finite element simulation (inset) shows that the maximum tensile stress of the cantilever beam is 9.08 GPa. **c** Top view SEM micrographs of cube corner indents performed with 150 mN and 500 mN load on WN$_x$/TiN superlattices. The lower part shows postmortem TEM observations after nano-indentation with a 500 mN load. **d** Comparison of hardness–toughness relationships for understoichiometric SL (WN$_x$/TiN and MoN$_{0.5}$/TiN) and other TMNs (transition metal nitrides, see the hardness–toughness data in Supplementary Table 1).

hexagonal)—although the crystal is macroscopically fcc—and become easier in the presence of vacancies, which change local environments (coordination, bond lengths, and angles). Here, the fcc structure is stabilized by the template effect of fcc TiN. A similar strategy may be applied to other group IV-VI transition metal nitrides that are meta-stable in the fcc phase, with several lower-symmetry (meta)stable phases available.

### Mechanical properties

Mechanical properties of the WN$_x$ ($x = 0.5$)/TiN SL were analyzed through bending and nano-indentation tests. Single cantilever bending experiments (without a pre-notch) of freestanding SL (Fig. 3a) revealed an astonishing 2.84% (±0.133%) bending strain (Fig. 3b). Besides, our SL coatings also exhibit a 9.08 GPa flexural strength (confirmed by the finite element simulation) and 302 GPa elastic modulus (finite element simulation, 299.6 ± 12.7 GPa calculated by the Euler-Bernoulli formula). Compared with the previous cantilever beam bending experiment (without pre-notch), the elasticity, strength, and elastic modulus here are significantly higher than the previous TMN material[17–23].

Furthermore, nano-indentation with a 150 mN load (Fig. 3c) shows a significant pile-up along the edges of the imprints, a clear sign of plastic flow under the compressive loading conditions during cube-corner indentation. Postmortem observation after the nano-

indentation with a 500 mN load proves plastic deformation in the forms of bending and compression in WN$_x$/TiN SL. By fitting the curvature of the bending layers and comparing them to the as-deposited SL, the current indentation deformation is shown to achieve ~6% plastic bending strains and ~22% local plastic compressive strains (near the impression tip region, Supplementary Fig. 8). Therefore, WN$_x$/TiN SL exhibits certain plastic deformation ability under compressive loading.

The WN$_x$/TiN SL also presents high hardness (~37 GPa) and fracture toughness. The cantilever beam test shows WN$_x$/TiN (5 nm/5 nm bilayer thickness, current observed sample) fracture toughness can be as high as 4.7 MPa$\sqrt{m}$[16]. Compared with other stoichiometric nitrides (Fig. 3d, Supplementary Table 1), the intrinsic fracture toughness ($K_{ic}$) of such understoichiometric SL system[24] is almost the highest among TMN systems[25], and twice the value measured for single-crystal TiN. On the atomic scale, toughening effects in understoichiometric WN$_x$ SL were predicted by DFT ductility indicators, bulk-to-shear modulus (B/G) ratio, and Cauchy pressure (CP). In particular, DFT results show a remarkably high B/G together with significantly high and positive CP for WN$_x$ ($x = 0.5$)/M*N or WN$_x$ ($x = 0.5$)/M*C (M*=Nb, Ti, Ta, Zr, Hf) SLs (Supplementary Table 2). These values are indicative of improved metallic bonding character, which can enhance the ductile response upon loading[14,16,24].

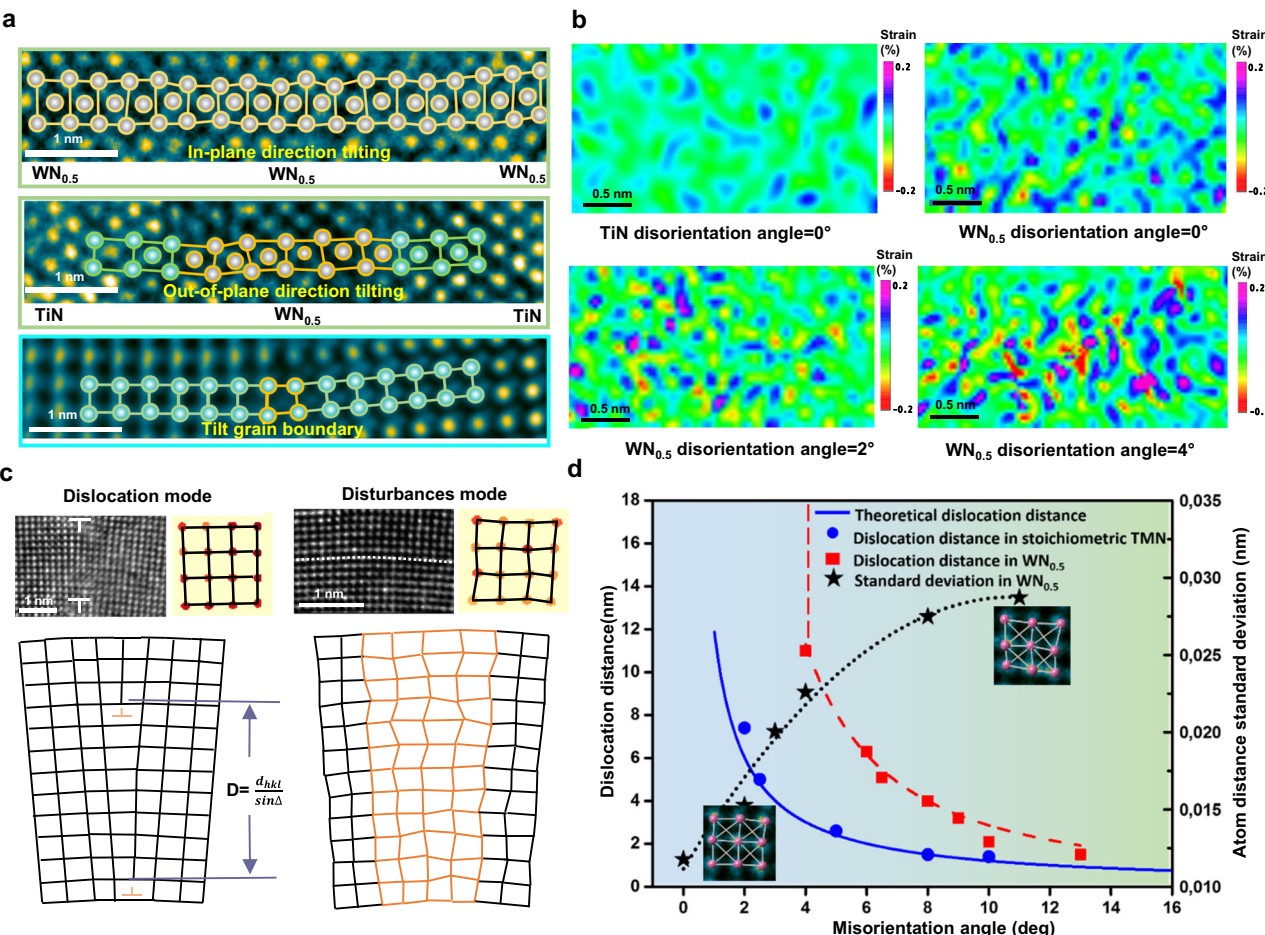

**Fig. 4 | Plastic deformation mediated by unit-cell disturbances. a** HRTEM observation of the atomic structures of the lattice rotation region in unstoichiometric TMN (in-plane and out-of-plane direction tilting in $WN_x$ layer) and stoichiometric TMN (i.e., $Ti_{0.67}Al_{0.33}N$, see Fig. S11), respectively. Atoms marked in yellow show severe unit cell distortions. **b** Geometric phase analysis results of TiN (misorientation angle = 0°) and $WN_x$ layer under different misorientation angles (~0°, ~2° and ~4°). The results showing the shear strain mapping ($g_1 = (11\bar{1})$/$g_2 = (1\bar{1}\bar{1})$), where the spatial resolution is about 3 Å. A misorientation angle of 0° is taken from as-deposited samples. **c** Lattice rotation is mediated by different defect types. For stoichiometric TMN, its lattice rotation will be accomplished by dislocation arrays on grain boundaries (see HRTEM observations on the left). For unstoichiometric TMN $WN_x$, its lattice rotation will be facilitated by unit-cell disturbances (see HRTEM on the right). The fitting results of the atomic positions are shown in the adjacent parts, which clearly show unit-cell disturbances in $WN_x$. Two schematic diagrams showing dislocation and unit-cell disturbances mediated lattice rotation, respectively. **d** Dislocation distance and unit-cell disturbances (standard deviation of atomic distances) as a function of tilting angles. The dislocation distances of the red and blue points are statistics from $WN_x$ and stoichiometric TMN, respectively. The solid blue curve is derived from the Frank-Bilby equation. The dotted line shows a fit to measured data. The standard deviation of atomic distances is derived from HRTEM observations with different misorientation angles (see Supplementary Fig. 15, Fig. 16).

## Deformation mechanism

Postmortem TEM observations from nanoindentation confirmed the plastic deformability of the $WN_x$/TiN superlattice. We note from images of the indented sample (Supplementary Fig. 9) that the SL bending occurs via continuous and gradual lattice rotation at a slight angle (about 1.3°/100 nm, Supplementary Fig. 9). The lattice rotation angle is almost equal to the macroscopic bending angle (see Supplementary Fig. 9a for the definition). Since interfacial plasticity does not generate crystal orientation variation[26], we assume that the bending mainly occurs by intra-layer plasticity. According to dislocation theory, a large number of geometrically necessary dislocations (GNDs) will be adapted to realize the lattice rotation. For ~6% plastic bending SL (indented sample, Fig. 3c), dislocation theory requires a GND density increase by $2.03 \times 10^{13}/cm^2$. The actual dislocation density after deformation, however, increases barely by ~$0.31 \times 10^{13}/cm^2$ (Supplementary Fig. 10). Thus, the increased dislocation density is about 15% of the theoretical value, contributing approximately 1% to the bending strain.

How does $WN_x$/TiN SL actually deform, then? To propose an explanation, we comprehensively analyzed atomic resolution images with various misorientations (here, misorientation is defined as the maximum lattice angle difference observed in the HRTEM images) features taken from the impression surface regions in the indented samples. For stoichiometric TMN deformation (Supplementary Fig. 11), lattice rotation is usually accomplished by forming a grain boundary (GB). However, the lattice rotation in the $WN_x$ layer (Fig. 4a, including in-plane and out-of-plane direction tilting) is different from that in the stoichiometric case (the bottom image in Fig. 4a and Supplementary Fig. 11). In contrast, the main features of lattice rotation in $WN_x$ are: (i) $WN_x$ layer does not have the apparent grain boundary structure for altering the orientation; (ii) the unit-cells display further distortions near the equilibrium positions of the fcc lattice, i.e., unit-cell disturbances.

Here, we mapped the varying degree of such disturbances by the geometrical phase analysis (GPA, Fig. 4b, and corresponding raw HRTEM images shown in Supplementary Fig. 12). When reasonable

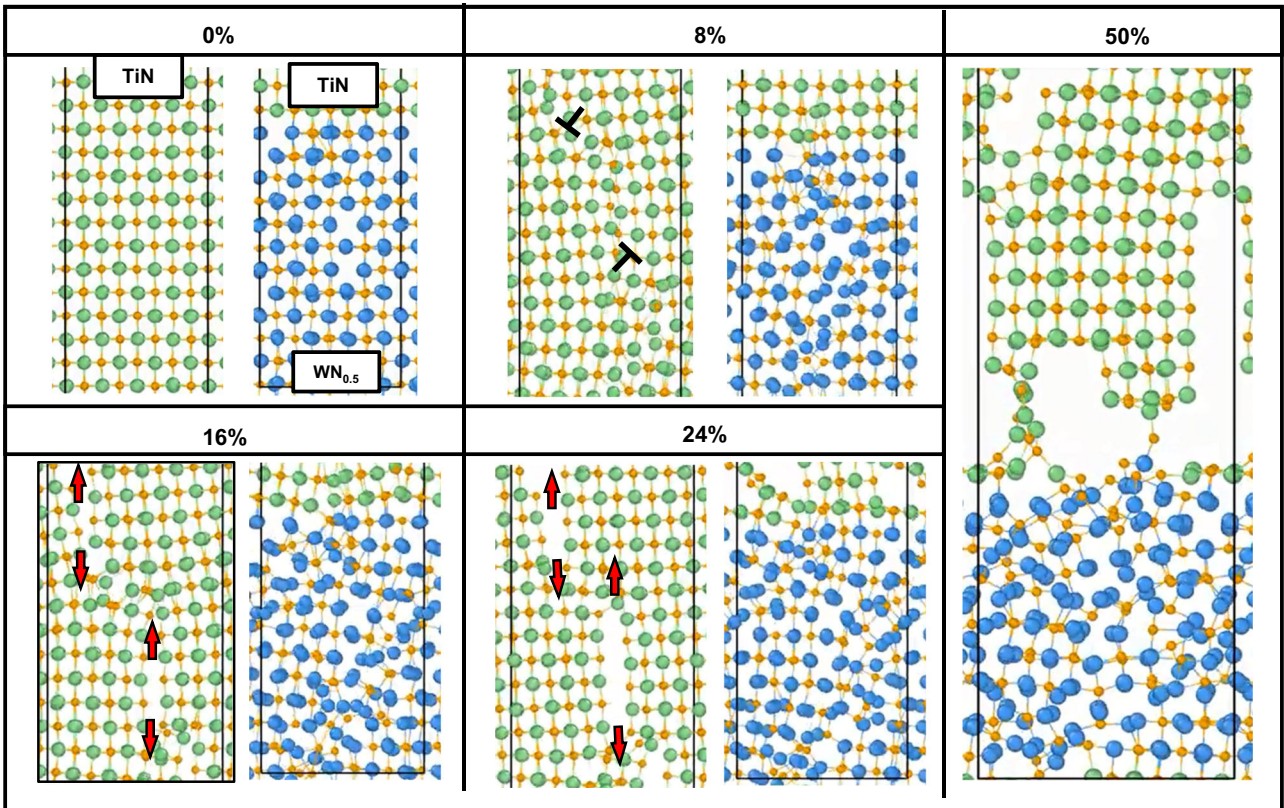

**Fig. 5 | AIMD simulated TiN and WN$_x$/TiN deformation.** AIMD (ab initio molecular dynamics) local magnification snapshots of monolithic TiN and WN$_x$/TiN SL (superlattice, ≈5 nm bilayer thickness) at selected strain steps during <110> tensile tests (in-plane elongation). Red arrows indicate crack initiation and propagation near dislocations in monolithic TiN and SL.

spatial resolution is utilized, local unit-cell scale distortions can be visualized instead of the usual nanoscale lattice strain. By comparing the strain maps obtained (Fig. 4b), we found that WN$_x$ (misorientation 0°) exhibits a higher unit-cell distortion than TiN. And the degree of unit-cell distortions in WN$_x$ increases significantly with the lattice rotation (i.e., misorientation 2° and 4°).

Conventionally, a small tilt GB is composed of an array of spaced edge dislocations to relax lattice strain (Fig. 4c). According to the Frank-Bilby equation[27,28], the distance between dislocations is a function of the misorientation angle, i.e., $d = \frac{b}{\Delta} = \frac{d_{hkl}}{\sin\Delta}$, where $b$ is Burgers vector, $\Delta$ is misorientation angle, and $d$ is interplanar spacing. However, for the case of small misorientation angles in WN$_x$, unit-cell disturbances mediated lattice rotation can be a dislocation-free mode. GPA results (Fig. 4b) and atomic-level observations (Supplementary Fig. 13, 1.5°, and 3.5° misorientation) indicate that dislocations are largely suppressed when the misorientation angle is less than 4°. Microscopically, the entire bending of the coating is realized by continuous and gradual lattice rotation at a tiny angle (Supplementary Fig. 9). In such a case, the strong suppression of deformation dislocations in WN$_x$ is due to significant unit-cell disturbances (Fig. 4c, disturbance mode).

For a larger misorientation angle (>4°), we performed extensive HRTEM-based statistical analyses. Our observations unveil that the Frank-Bilby equation still holds for the stoichiometric TMN (Fig. 4d, blue curve). However, the dislocation distance greatly expands when the misorientation angle is 4°–11° in the WN$_x$ layer (Supplementary Fig. 14). The measured dislocation distance (Fig. 4d, red square) is above the theoretical prediction (Fig. 4d, solid blue curve), invalidating the Frank-Bilby equation. To track the origin, the distortions were determined by quantitative measurements (Supplementary Figs. 15, 16, and 17). We further found that the extent of unit-cell

distortion increases with the misorientation angle (Fig. 4d, star shape). This discloses an intrinsic connection between unit-cell distortion and dislocation distance. Taken together, we conclude that vacancy-induced unit-cell disturbances substitute (or partially substitute) dislocations as carriers of plastic deformation. Such disturbance behavior adapts the deformation via unit cell volume and atomic density variations, different from current plastic deformation manners in TMN[29–33].

### Deformation behavior during AIMD tensile tests

We performed ab initio molecular dynamics (AIMD) simulations at 300 K to underpin the observed dominating role of unit-cell disturbances. To gain insights into the plastic behavior of the WN$_x$ ($x = 0.5$)/TiN SL, we carried out AIMD simulations of tensile deformation along <110> and <100> directions (Supplementary movie 1, movie 2, Fig. Supplementary Fig. 18), respectively. A series of snapshots illustrate the key steps of the <110> deformation process (Fig. 5, Supplementary movie 1). For stoichiometric monolithic TiN, dislocation nucleation is observed at ($\varepsilon = 8\%$), which is hardly visible in SL WN$_x$. Upon further <110> strain increase, Ti-N bond breakage initiates a void opening orthogonal to the loading direction ($\varepsilon = 16\%$). Higher <110> strain ($\varepsilon = 24\%$) induces the growth of a few voids in the TiN layer of the SL (close to the interface). Nonetheless, these voids are relatively small compared to those in the monolithic TiN.

AIMD simulations indicate that unit-cell disturbances in the WN$_x$ layer increase with increasing strain and are more severe near N vacancy sites. By introducing N vacancies, the bonding character changes to more metallic, i.e., is dominated by delocalized electrons that control the ability to redistribute stresses via plastic deformation (see a significant change in the W-W bond distribution in Supplementary Fig. 19 and 20). We expect that, similar to, e.g., VN$x$[13], the

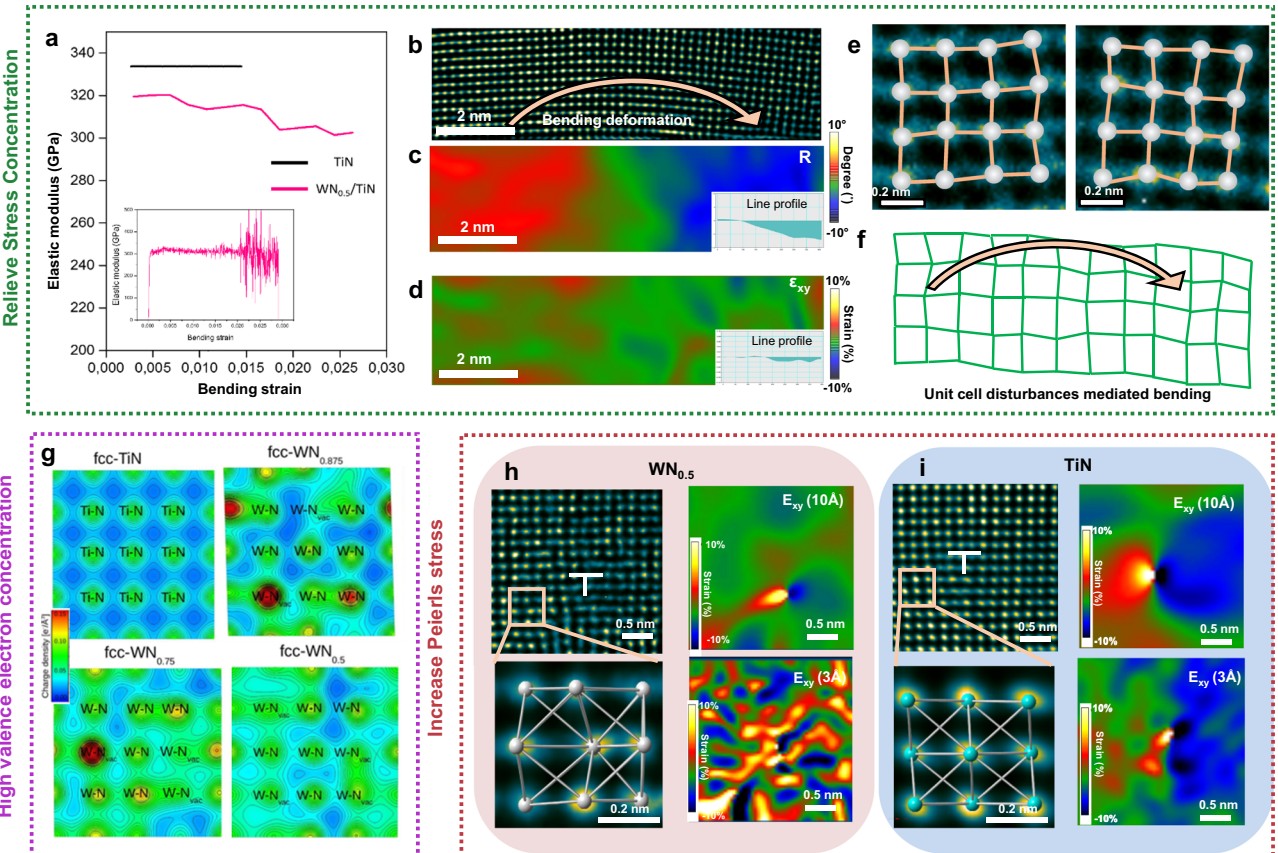

**Fig. 6 | Toughening and strengthening mechanism in WNₓ/TiN. a** Elastic modulus of WNₓ/TiN SL and monolithic TiN under different strains. The curve is derived from the derivative of the stress-strain curve for cantilever beam bending (for WNₓ/TiN SL, the derivative step is about 0.2% strain). Insert image showing the results of the derivative step is about 0.004% strain. **b** HRTEM image in WNₓ with a certain misorientation angle taken from the indented tip surface (150 mN). **c** GPA (Geometric phase analysis, rotation angle mapping) results of WNₓ indicate a maximum of 7° rotation angle (as shown in the inserted line profile). **d** GPA shear strain mapping indicates a maximum 1.8% shear strain (as shown in the inserted line profile). GPA results show the shear strain mapping ($g_1 = (100)/g_2 = (010)$), where the spatial resolution is about 10 Å. **e** Two enlarged HRTEM images (locally clipping images from B)) indicate WNₓ has different severe unit-cell distortions from place

to place. **f** Schematic diagram of distortion mediated deformation. **g** Qualitative comparison of electronic charge densities calculated using DFT for fcc-TiN and fcc-WNₓ (with different nitrogen contents). The visualization is along the (001) plane, i.e., between the first nearest neighbor Ti-N and W-N atoms. **h, i** HRTEM images show the intrinsic dislocation structure in WNₓ and TiN (both dislocations from the as-deposited SL), respectively. Inserted GPA results ($g_1 = (100)/g_2 = (010)$), from HRTEM (**h** and **i**) show the distribution of shear strain near the dislocations, where the spatial resolution is about 10 Å and 3 Å. For 3 Å spatial resolution, due to the higher spatial resolution chosen here, the GPA results show the lattice distortions instead of the long-range lattice strain. Enlarged HRTEM results show lattice distortions near the dislocation core. Comparatively, TiN exhibits uniform shear deformation, while the WNₓ lattice presents significant distortion.

metallic shear-sensitive d(W)−d(W) interactions increase. As the WNₓ layer at ~50% of tensile strain is still free of dislocations (Fig. 5), AIMD simulations support the experimentally observed anomalous dislocation density in WNₓ/TiN SL. Furthermore, AIMD underpins the experimental hypothesis on unit-cell disturbances in WNₓ being the main carrier of plastic deformation, where vacancy-induced unit-cell disturbances significantly improve the plastic deformation capability of WNₓ.

## Discussion

The improvement of ceramics' plasticity often comes at the expense of their hardness. For instance, the ceramic/metal multilayer system[20,34–36] achieves better deformability and fracture toughness, but these designs barely provide high strength. Introducing vacancies may reduce the theoretical strength of ceramics[13,14,37]. However, the WNₓ/TiN SL achieved high deformation ability while maintaining very high strength and toughness. We ascribe to three factors:

(i) With high vacancy concentrations (~50%) and a disordered distribution, the disturbance of the unit cell in WNₓ allows for the absorption of deformation energy. The cantilever beam bending experiment shows purely elastic deformation. However, upon closer

analysis of the stress-strain curve, we discovered that it exhibits nonlinear elastic deformation. Figure 6a shows the variation in elastic modulus, which exhibits a decay with strain and is accompanied by a slight fluctuation (the inserted figure, derived with a different derivative step of ~0.004%, exhibits severe oscillations). In contrast, the elastic modulus for TiN (a monolithic coating) does not present any decay nor fluctuation during the bending process.

The nonlinear elastic deformation typically arises from strain-induced phase transformation in shape memory alloys[38,39] or non-uniform stress distribution in porous materials[40]. Thermodynamically, these processes consume overall elastic deformation energy. Unlike the materials mentioned above, we believe the superlattice's nonlinear elastic deformation stems from unit-cell disturbances (induced by 50% vacancies) in WNₓ during deformation. Simulation of stoichiometric TiN subjected to [100] tensile elongation at 0 K (Supplementary Fig. 21) reveals elastic deformation characteristics, such as unchanged atomic arrangements and uniformly elongated (or shortened) nearest neighbor Ti-Ti, Ti-N, and N-N bonds. However, the equivalent elastic strains applied to WNₓ trigger strongly inhomogeneous unit cell distortions (as framed in Supplementary Fig. 21). Such local inhomogeneous response under load conditions leads to macroscopic-scale

nonlinear deformation. In other words, it behaves like a unit-cell scale "porous structure" to a certain extent.

Further experimental analysis confirmed the relaxation effect of unit-cell disturbances on lattice strain. The HRTEM image (Fig. 6b) in $WN_x$ with a certain misorientation angle was taken from the indented tip surface (150 mN). GPA results show the shear strain mapping ($g_1 = (100)/g_2 = (010)$), where the spatial resolution is about 10 Å. Traditionally, lattice rotation ($R^{max} = 7°$ in Fig. 6c) requires a certain elastic strain or geometrically necessary dislocations to achieve. However, GPA measurements (Fig. 6d) have barely shown lattice strain in $WN_x$ (the maximum lattice shear strain is only 1.8%), and no dislocations are present. These results suggest that most of the lattice strain can be relieved via unit-cell disturbances, as atomic-resolution images (Fig. 6e) display. Over a large scale, the collection of these local unit-cell disturbances could accommodate the macroscopic-scale deformation (Fig. 6f). In this way, we underline that a high density of disordered vacancies could lead to enhanced elastic and plastic deformability of the overall superlattice, as the experiment (Fig. 3a) and simulations (Fig. 5) corroborate. As a result, the high elastic deformation capacity of $TiN/WN_x$ leads to higher fracture energy than in TiN (see previous bending results for pre-notched cantilever beams[16]). Materials with high fracture energy can absorb more energy and exhibit better resistance to crack propagation. Our SEM micrographs (Supplementary Fig. 22) of fracture surfaces show stepped features. The fact that the fracture here is very rough, while the crack extends in a tortuous manner, suggests that the $WN_x$ SL can significantly deflect the crack path. The phenomenon can be explained by the unit-cell-disturbance mechanism proposed in this work. HRTEM observations and GPA analysis (as proved in Fig. 6d and Supplementary Fig. 23) show that unit cell disturbances can effectively attenuate the elastic stress field, which greatly avoids the stress concentration effect during elastic bending deformation. Since the stress field affects the crack propagation, the presence of $WN_x$ as a low-stress zone could deviate the crack from high-stress areas, i.e., crack deflection (Supplementary Fig. 22).

(ii) The presence of nitrogen vacancies effectively increases the electron concentration in the (remaining) metal-N bonds and metal-metal bonds[8,41,42]. For $WN_x$, increased electron accumulation is predicted not only between W and N atoms but also in the surrounding space (Fig. 6g), indicating electron delocalization and enhanced metallic-like character. Such electron delocalization in the vicinity of N-vacancies occurs since electrons cannot be transferred from W to N (ionic transfer due to N electronegativity). Thus, electronic delocalization facilitates the enhancement of plastic deformation ability, and intrinsic fracture toughness of $WN_x$ with increasing N vacancy content, up to 50% vacancies.

(iii) The presence of vacancies causes distortions in the unit cell, which hinders dislocation nucleation (as confirmed by the above section and Fig. 4) and motion. In the case of ultra-high concentration of disordered vacancies, distortions are present in almost every unit cell (as seen in Fig. 2). This widespread distortion leads to an overall altered atomic arrangement that introduces an additional energy barrier for dislocation nucleation.

On the other hand, $WN_x$ also increases the resistance to dislocation movement. Consequently, the $WN_x/TiN$ superlattice can approach the strength limit. Severe unit-cell distortions bring a high critical resolved shear stress to hinder dislocations from the neighboring layers crossing the $WN_x$. This can be demonstrated by comparing the core widths of dislocations[43] in the as-deposited film. GPA analysis shows that the scale of the elastic stress field of the dislocation core in $WN_x$ (Fig. 6h, as-deposited SL) is very small compared to TiN (Fig. 6i), i.e., $WN_x$ has a dense dislocation core. In contrast, TiN shows a non-dense dislocation core. The classical theory for the lattice friction stress (i.e., Peierls stress) can be expressed using the following

Equation[43]:

$$\sigma = 2MG/(1-\nu) \cdot \exp(-2\pi w/b) \qquad (1)$$

where $M$ is the Taylor factor, $G$ is the shear modulus, $\nu$ is the Poisson's ratio, $w$ is the width of a dislocation, and $b$ is the magnitude of the Burgers vector. The dense dislocation core (smaller $w$) in $WN_x$ leads to high lattice friction stress. Thus, the vacancies impede dislocation migration, inhibit dislocation nucleation, and increase Peierls stresses via dislocation-pinning[44], thereby suppressing slip-mediated deformation (no sliding deformation was observed after indentation in the current coating) and improving strength upon compression, e.g., hardness in nanoindentation.

In conclusion, through atomic-resolution TEM observations and quantitative analysis, we clearly demonstrate the unique structure of $WN_x$, which has ultra-high concentration of disordered N vacancies (up to 50%) and exhibits severe unit-cell distortions. Microcantilever tests show that the elasticity (~2.8%) and flexural strength (~9 GPa) of $WN_x/TiN$ SL exceed those of the stoichiometric metal nitrides system. The enhanced mechanical properties of $WN_x/TiN$ SL can be attributed to effects induced by the ultra-high concentration of vacancies: unit-cell disturbances that absorb deformation energy, increased valence electron concentration, and hindered dislocation nucleation/motion. Our study underscores the importance of rational vacancy structure design, i.e., characterized by high vacancy concentration and disordered distribution, which can improve the deformation ability and strength of ceramics.

## Methods
### Sample preparation
All superlattice and monolithic TiN were synthesized via unbalanced DC reactive using an AJA International Orion 5 magnetron sputtering deposition system. For $WN_{0.5}/TiN$ SL (~1.4 μm total film thickness), one 3′. Ti target and one 2′. W target were placed onto the respective cathodes and used the computer-controlled shutter system to synthesize SL coatings with certain bilayer periods (TiN ~5.1 nm, $WN_{0.5}$ ~ 5.1 nm). Previous studies[16] show that different flow ratios affect the vacancy concentration and crystallographic structure of $WN_x$. Here, to obtain $WN_{0.5}/TiN$ SL with ideal quality, the reactive magnetron sputtering process was carried out at 500 °C (substrate temperature) in an $Ar/N_2$ mixed gas atmosphere with a total pressure of 0.4 Pa and an $Ar/N_2$ flow ratio of 5.3 sccm/4.7 sccm. To avoid the intermixing of the two-layer materials via excessive ion bombardment, we applied a rather low bias potential of −40 V to the MgO (100) substrate, just enough to obtain a dense coating morphology.

The TiN/AlN superlattice thin film (representative of stoichiometric coatings, Fig. 4a) is with ~1.6 μm total film thickness, with a bilayer thickness of 2.5 nm (TiN ~1.7 nm, AlN ~0.8 nm). The reactive magnetron sputtering process was carried out at 700 °C (substrate temperature) in an $Ar/N_2$ mixed gas atmosphere with a total pressure of 0.4 Pa and an $Ar/N_2$ flow ratio of 7 sccm/3 sccm. The bias potential is −40 V. Detailed TiN/AlN and $WN_{0.5}/TiN$ coating preparation can be found elsewhere[16,45].

### Nanomechanical testing
The bending strain/stress of $WN_x/TiN$ thin films was unveiled by performing single cantilever bending experiments of freestanding coating material. These cantilevers were prepared with a focused ion beam (FIB) workstation (FEI Quanta 200 3D FIB). These cavities below the coating were cut using subsequent steps of 5 nA, 3 nA, and 500 pA. The cantilevers themselves were cut with 1 nA to obtain the rough shape and 500 pA for fine patterning steps. Cantilever dimensions are ~13 μm/1.4 μm/3 μm. The bending tests were executed inside the

aforementioned FEG-SEM by a PI85 PicoIndenter (Hysitron). The spherical diamond tip attached to the indenter had a tip radius of about 1 μm. The loading speed is 20 nm/s. The bending stress was calculated according to:

$$\sigma = \frac{6P \cdot L}{B \cdot t^2} \tag{2}$$

$$\varepsilon = \frac{3t \cdot w}{2L^2} \tag{3}$$

$$E = \frac{4}{B} \times \frac{dp}{dw} \times \left(\frac{L}{t}\right)^3 \tag{4}$$

where $P$ and $w$ are the load and deflection of the cantilever during loading. $B$, $L$, and $t$ are the width, length, and thickness of the cantilever, respectively.

The nanoindentation was performed with an Ultra Micro Indentation System (UMIS, Fischer-Cripps Laboratories) equipped with a cube corner diamond tip using a maximum load of 150 mN and 500 mN. The FIB cutting position is chosen near the indenter tip, and cutting is along the <110> or <100> direction of the SL. For TEM samples, we deposited a W protective layer. The rough milling current is set at 1 nA (30 keV) to prepare two trenches of a certain depth. The sample thinning undergoes rough and fine polishing with currents of 500 pA (30 keV) and 50 pA (30 keV), respectively. Fine polishing is conducted after transferring the sample to a copper grid.

## HRTEM characterization
A 200 kV field emission TEM (JEOL 2100F) equipped with an image-side $C_S$-corrector was used in the high-resolution TEM (HRTEM) study, which demonstrates a resolution of 1.2 Å at 200 kV. The aberration coefficient is set close to zero under which the HRTEM images were taken under slightly over-focus conditions (close to the Scherzer defocus). A CCD Orius camera was used to record HRTEM images, where image sizes are 2048 pixels × 1336 pixels. HRTEM image simulations were executed using the JEMS software multi-slice approach (copyright P.|A. Stadelmann, EPFL, Switzerland). The simulated model of N vacancies is derived from the un-relaxed model of DFT calculations. The simulated model of lattice distortion is derived from the post-relaxation model of DFT calculations.

The atom distances were calculated by determining atomic positions which were fitted by a spherical Gaussian function using an algorithm in Matlab script[46]. The signal intensity distributions of the W-site and N-site were determined via Atomap[47], with analysis being performed with custom Python scripts.

The strain fields in $WN_x$ or TiN were calculated based on the $C_S$-corrected HRTEM images by the geometric phase analysis (GPA) method. According to the GPA algorithm, the displacement fields can be obtained by selecting two non-collinear Bragg vectors in the power spectrum generated from a high-resolution TEM image. Here, we choose different Mask sizes to obtain the maps with different spatial resolutions (0.3–1 nm).

For the dislocation density determination in the as-deposited state and deformed state (post-mortem observation after nanoindentation) $WN_x/TiN$, a total of 60 (30 per state) HRTEM images of different regions (recorded at 600-800KX) are counted. All statistical dislocations are 1/3{111} Frank dislocations observed under the fcc <110> zone axis.

## STEM/EELS/EDAX characterization
STEM-EELS investigations were performed in a probe-corrected FEI Titan$^3$ G2 60–300, operated at 300 kV. The microscope is equipped with a Gatan Imaging Filter (GIF) Quantum with a Gatan K2 Summit direct-electro-detection camera. Spectrum images are acquired with a convergence angle of 19. 6 mrad and a collection angle of 25 mrad, with a dispersion of 0.1 eV/ch with a pixel size of 0.4 A and a dwell time of 10 ms per pixel for 300 × 200 pixels. Spectra in Fig. 1A are summed over 45 * 25 pixels.

For Fig.S10, a 300 kV field emission TEM (JEOL ARM300F) equipped with a double $C_S$-corrector was used in this study to acquire the high-spatial elemental maps. Two windowless Energy-dispersive X-ray spectroscopy (EDXS) detectors, each of which has an active area of 100 $mm^2$, are equipped on the microscope, which is very close to the specimen with a high solid angle (1.7 sr).

## Finite element simulation
A 3D finite element model of a cantilever was established within the commercial code Abaqus. This cantilever had a length of 13.3 μm, a width of 2.9 μm, and a height of 1.55 μm. The right end of the cantilever was fixed and with finer mesh in the length direction of the cantilever (the minimum edge length of one element was 1 nm). With distance from the fixed end, the element edge length in the length direction of the cantilever gradually increased up to 0.5 μm. The mesh size at the cross section was around 81 nm. This operation yielded 49335 nodes and 43296 linear hexahedral elements.

The displacement control was applied at the right end in the vertical direction with a contact area of 133 nm × 2.9 μm to represent the contact surface between the cantilever and the indenter. A displacement rate of 50 nm/s was considered. Only the elastic material behavior of ceramics was taken into account. A constant Poisson ratio of 0.39 was defined, whilst the Young's modulus was modified to fit the linear part until the peak load of an experimental curve received from nano-indentation.

## Ab initio density functional theory and ab initio molecular dynamics calculations
Ab initio DFT calculations were carried employing the Vienna Ab-initio simulation Package (VASP)[48,49] together with plane-wave projector augmented wave (PAW) pseudopotentials[50] and the Perdew–Burke–Ernzerhof generalized gradient approximation[51]. Cubic fcc supercells (space group Fm-3m, 64 atoms) of TiN and WN were relaxed, with the plane-wave cutoff of 600 eV and the reciprocal space sampling with Monkhorst-Pack meshes[52] ensuring a total energy accuracy of $10^{-4}$ eV/at. To model substoichiometric $WN_x$ structures, $N$ vacancies (12.5, 25, and 50 at%) were generated in a disordered manner following the Special Quasirandom Structure method[53] and the supercells were fully relaxed in terms of volume, shape, and atomic positions. Tensile loading along the [100] axis was simulated by sequentially elongating the supercell (with a 2% strain step), allowing for relaxations (due to Poisson's contraction) in the [010] and [001] directions. The radial distribution function was evaluated using the OVITO package[54].

AIMD simulations were performed using VASP. A 576-atom fcc-based supercell, with 288 metal and 288 nitrogen atoms and total dimensions of ≈ (0.9 × 1.2 × 5) $nm^3$, served as a model of TiN. The TiN/$WN_{0.5}$ SL model with a bilayer period of ≈5 nm was built by replacing half of the Ti atoms by W (keeping the [001] direction orthogonal to interfaces) and randomly distributing anion vacancies in the $WN_x$ layer. Room temperature equilibrium structural parameters of monolithic TiN and TiN/$WN_{0.5}$ SL were optimized using NPT sampling of the configurational space (Parrinello-Rahman barostat[55] and Langevin thermostat set to 300 K). The computational procedure for modeling in-plane (parallel-to-interfaces) [100] and [110] tensile deformation followed ref. 56.

## Data availability
The data that support the findings of this study are available from the corresponding author upon request.

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

## Acknowledgements

The authors thank Dr. J. Buchinger for the thin film synthesis. Dr. Matthias Bartosik is acknowledged for his initial discussion of experiments. The authors thank Prof. Jian Wang (Department of Mechanical and Materials of Engineering, University of Nebraska-Lincoln) for the helpful discussion about interpreting mechanical properties and deformation mechanisms. This work is financially supported by FWF P 33696 (Z.C., Y.H., Z.Z.). Z.G. thanks the China Scholarship Council (CSC, 201908440933) for the support. D.G.S. gratefully acknowledges financial support from the Competence Center Functional Nanoscale Materials (FunMat-II) (Vinnova Grant No. 2022-03071) and the Swedish Research Council (VR) through Grant N° VR-2021-04426. Calculations and simulations were performed using resources provided by the Swedish National Infrastructure for Computing (SNIC), partially funded by the Swedish Research Council through Grant Agreement N° VR-2015-04630.

## Author contributions

Z.C. conducted the TEM/HRTEM experiments, designed the experiments, and wrote the manuscript with input from all authors. Z.C. and Y.H. analyzed data. N.K. and D.S. performed the computational studies. S.F. prepares TEM samples. G.H. and G.K. perform STEM/EELS/EDXS experiments. S.J. performed FE simulation, and Z.G. and P.M. performed Nanomechanical testing and thin film synthesis. Z.Z. conceived the idea, supervised the entire project, and revised the manuscript.

## Competing interests

The authors declare no competing interests.
