## [Peer Review File · Nature Communications]

REVIEWER COMMENTS

Reviewer #1 (Remarks to the Author):

The authors proposed unit-cell disturbances with a very high content of disordered nitrogen vacancies in a superlattice nitride ceramic that would induce atomic-level distortions to relieve the high stress concentration upon mechanical loading and enhance the deformability and toughness of the brittle covalent material. This is a comprehensive study with very detailed investigations and advanced characterization as well as delicate atomistic simulations as strong evidences of the proposed vacancy-disturbed structure and the consequent deformation mechanism, being worth a publication in a renowned journal. However, some major concerns given below are suggested to be carefully considered and addressed:

(1) There is no doubt of the lack of N atoms at these lattice sites. But, since the number of N atoms is much lower than (only 50% of) the number in a stoichiometric FCC nitride, how would the authors define the structure as “vacancies” in an NaCl-type FCC WN_x lattice than N interstitials at the face- or edge-center of a BCC W unit cell. How would the authors expect the lattice constant of a regular BCC W with N interstitials to be, also close to the size of an FCC WN_x with vacancies? A W-W metallic bonding structure with interstitials is believed to be much more ductile than a brittle W-N bonding structure with vacancies.

(2) The bending strain is different from the normal strain and strongly depends on the geometry of the specimen. But, it reads like the authors implied a correlation of bending strain to toughness. As not a notch was made on the specimens, even a diamond nanoneedle could be superelastic (Science 360 (2018) 300). Even just in comparison of different samples, the high elastic bending strain doesn't correspond to high fracture toughness (particular the KIC). Since the data of fracture toughness were published in previous studies, the statements are suggested to be more focused on the deformation mechanism, i.e. the “vacancy-mediated” toughness, than the “toughness”. Moreover, did the bent cantilever elastically recover, or with any residual plastic deformation? How would the authors comment on it from the perspective of elastic or plastic deformation?

(3) A high local plastic strain of about 22% was observed in the indentations with a very low density of dislocations, but the strain might include not only plastic deformation but also the microcracking of the film or the delimitation (or shear) at the interface, as typically observed in laminated ceramic coatings also with a low density of dislocations. How would the authors expect or observe? Has any similar evidence been seen in the bending tests? Or, would the authors consider to verify the high plastic strain by using micropillar compression?

(4) Most importantly, the key point of this work is the enhanced toughness and improved plasticity through the function of the vacancy-caused lattice disturbances at a low dislocation density. It might be possible, and all the discussion is good (though the latter part is a little bit too long). But, these disturbances look immobile. The strain rate of typical crystalline materials can be estimated by the Orowan equation. If it is feasible that the atomic-level vacancy-caused unit-cell disturbances are the carriers of large-scale plasticity, could the authors estimate the density and mobility of these defects? Since atom arrangements are necessary for plastic deformation, for a very slight increase in the density of dislocations, what amount and what scale of the short-range displacements would contribute towards the total large plastic strain?

(5) Additionally, Figure 1c about the EELS spectra should be 1b. And, some characters in the figures are too small to be clearly seen.

Reviewer #2 (Remarks to the Author):

In the manuscript NCOMMS-23-29233 authored by Chen et al., the atomistic microstructure of vacancy-mediated superlattice (SL) W_{N_x}/TiN thin films is investigated, with a focus on its correlation with the mechanical properties. The study showcases cutting-edge experimental work, complemented by the implementation of ab initio molecular dynamics (AIMD) calculations. The work main contribution is the proposal of a novel deformation mechanism called "unit-cell disturbances." This mechanism is expected to show potential to enhance the elasticity, flexural strength, and fracture toughness of non-stoichiometric nanoceramics. This is of utmost importance, as it presents a new pathway to tackle the longstanding strength and toughness tradeoff often observed in brittle materials. The analytical transmission electron microscopy (TEM) work conducted in this study deserves special praise. Particularly remarkable is the observation of vacancies along $[100]$ and $[110]$ directions, revealing an exquisite microstructural insight.

I find the main manuscript idea very progressive, particularly the concept of plastic deformation being mediated by unit-cell disturbances, as depicted in the schematic illustration of Fig. 4c. The authors propose that critical stress concentrations can be alleviated through localized unit cell deformations, leading to improved deformability and toughness at the macro-scale. The notion of creating a cellular material with atomistic voids, where the bonding character changes to a metallic one, is quite captivating. This innovative concept not only offers promising insights into studying the mechanical properties of non-stoichiometric "atomistic composites" but also opens up avenues to explore their other physical properties. I am particularly interested in the potential of this approach to study materials that lie at the threshold between dislocation- and unit-cell disturbance-mediated plasticity.

Regarding the Editor's inquiries, it is to note that the research group has previously published numerous works investigating the correlation between mechanical properties and microstructure of nano-laminates. Notably, Buchinger et al.'s paper [19] also demonstrated similar effects, showcasing enhanced mechanical properties through the use of superlattice (SL) TiN/WN architectures. The key differentiating factor in NCOMMS-23-29233, compared to the previous works by other authors, lies in the introduction of the concept of unit-cell disturbances. This approach sets the current manuscript apart, offering mainly a perspective to interpret the mechanical properties of nano-laminates with a large concentration of vacancies.

The observed hardness of ~ 37 GPa and fracture toughness of ~ 4.7 MPa $\sqrt{\text{m}}$ reported in NCOMMS-23-29233 is comparable with the values reported already in the previous works of the authors, like 36.7 GPa and 4.6 MPa $\sqrt{\text{m}}$ in Ref. [19], respectively, as well as by others in similar materials.

The concept of improved mechanical properties (by using element deficiency and excess) within non stoichiometric nano-laminates is similar to that reported for self-assembled coherent cubic AlN/TiN [<https://doi.org/10.1016/j.actamat.2017.04.009>] for which also similar hardness and toughness values of 36 GPa and 4.7 MPa $\sqrt{\text{m}}$, respectively, were observed [<https://doi.org/10.1039/C8NR10339A>].

So, the manuscript's primary contribution lies in the introduction of a new deformation concept, rather than presenting a new material type with a unique microstructure or mechanical properties.

I have three concerns related to the manuscript main statement of unit cell disturbance-mediated deformation:

A) The authors state that “by introducing N vacancies, the bonding character changes to more metallic, i.e., is dominated by delocalized electrons that control the ability to redistribute stresses via plastic deformation”. The plastic deformation means for me an irreversible change of the sample shape during deformation. When performing bending tests on the micro-cantilevers, did the authors observe irreversible change in the cantilever shape (or the cantilever shape fully recovered after unloading)? The load-deflection curves in Fig. 3b indicate linear-elastic response without any “macroscopic” plasticity. The pile-ups in Fig. 3c could have also other origins than plastic flow.

B) While the authors introduce a novel deformation mechanism to explain the improved elastic deformability observed in Fig. 3a,c, there remains however uncertainty about the fracture behavior of WN_x/TiN , particularly the mechanism behind the linear-elastic fracture depicted in Fig. 3b. The nature of the fracture, whether it is transgranular or intercrystalline, is not entirely clear, especially considering that typically, TMN thin films tend to suffer from brittle fracture along grain boundaries or 2D-3D defects. Therefore, it raises questions about whether the WN_x/TiN nanolaminate used in this study is indeed monocrystalline, free from 2D-3D defects. It would be beneficial if the revised manuscript

includes a discussion on the expected fracture mechanism and a thorough analysis of the fracture surfaces of the cantilevers. It might be helpful to explore the presence of cleavage planes to gain further insights. Furthermore, addressing the preparation process of the monocrystalline W_Nx/TiN on MgO and discussing potential 2D and 3D structural defects (using conventional TEM?) would contribute to a more comprehensive understanding of the material's mechanical behavior. By providing additional clarity on the fracture mechanism and analyzing fracture surfaces, the authors could enhance the significance and reliability of their findings.

C) It would be also interesting to analyze mechanical properties of nonstoichiometric W_Nx , as a reference material, and compare them with the performance of the W_Nx/TiN nanolaminate. Currently, it is not clear how the synergetic effect of the SL and the vacancies actually controls and influences the fracture. It would be genius to compare the present material with W_Nx of high concentration of N vacancies to see the difference in the mechanical performance.

NC Inquiries:

- The correlation of micro-mechanical studies, analytical TEM and AIMD calculations is really nice and represents state of the art.
- The novelty of the work resides exclusively in introducing the concept of vacancies-mediated unit cell disturbance, which is stated to be responsible for better deformability of the nanolaminate. It can be expected that the work will attract the attention of the TMN community and stimulate new research. It appears that the work does not introduce a new material concept – non-stoichiometric superlattices and/or multilayered TMN microstructures with practically equal mechanical properties were already reported.
- Some statements, especially those related to the observed plastic deformation and linear-elastic fracture seems to be not justified. The authors could provide experimental evidence that the proposed unit cell disturbance mechanism improves also the fracture behavior of the nanolaminates – see the discussion (A-C) above.
- The used methodology is sound and the group work represents the state of the art in the field.
- The method description is sufficient and can be reproduced.

Reviewer #3 (Remarks to the Author):

The authors elaborated the vacancies-mediated mechanical property enhancement for W_Nx/TiN superlattices. The structural, compositional and nanomechanical analysis of W_Nx/TiN SL architecture is clear and convincing. Therefore, I recommend the paper to be published after the following comments are taken into consideration:

1. Fig. 1 (b) is mis-labelled as (c) for the EELS results.
2. In row #192-193, it is mentioned that the actual dislocation density after deformation increases by $0.31 \cdot 10^{13}/\text{cm}^2$ which is much smaller than GND increase of $2.03 \cdot 10^{13}/\text{cm}^2$. But in the supplementary material Fig. S9, it is written that GND will increase $0.203 \cdot 10^{12}/\text{cm}^2$, which is much smaller than actual measurement. Which one is correct? If the supplementary material is correct, then the plasticity can be achieved solely by dislocation generation/motion.
3. Could you explain why the unit-cell distortion will hinder the dislocation nucleation? If unit-cell is distortion energy is much higher than dislocation formation energy, will dislocation forms instead of unit cell distortion?

REVIEWER COMMENTS

Reviewer #1 (Remarks to the Author):

The authors proposed unit-cell disturbances with a very high content of disordered nitrogen vacancies in a superlattice nitride ceramic that would induce atomic-level distortions to relieve the high stress concentration upon mechanical loading and enhance the deformability and toughness of the brittle covalent material. This is a comprehensive study with very detailed investigations and advanced characterization as well as delicate atomistic simulations as strong evidences of the proposed vacancy-disturbed structure and the consequent deformation mechanism, being worth a publication in a renowned journal. However, some major concerns given below are suggested to be carefully considered and addressed:

(1) There is no doubt of the lack of N atoms at these lattice sites. But, since the number of N atoms is much lower than (only 50% of) the number in a stoichiometric FCC nitride, how would the authors define the structure as "vacancies" in an NaCl-type FCC WN_x lattice than N interstitials at the face- or edge-center of a BCC W unit cell. How would the authors expect the lattice constant of a regular BCC W with N interstitials to be, also close to the size of an FCC WN_x with vacancies? A W-W metallic bonding structure with interstitials is believed to be much more ductile than a brittle W-N bonding structure with vacancies.

Thank the reviewer for the excellent comments.

Within the WN_x layer, the W atoms occupy a face-centered cubic (FCC) arrangement, while N atoms (along with their vacancies) are situated on an interpenetrating FCC lattice.

The reason why we define the current structure as "vacancies" in a NaCl-type FCC WN_x lattice rather than N interstitials at the face- or edge-center of a BCC W unit cell is as follows:

- (a) All the experimentally measured results point to the FCC WN_x lattice (see the following), i.e., Selected area electron diffraction (SAED), HRTEM/STEM, and XRD results. And the calculations also points out that WN_x could accommodate a large amount of N vacancies while maintaining the stable FCC lattice but with slight different lattice constants.
- (b) In fact, metal BCC-W could contain a marginal of nitrogen atoms as interstitials[1], consequently improving the mechanical properties. However, only a few atom percent (~ at 5%) is detected, which is way lower than that in the current WN_x structure (even for 50% vacancies). As such a structure retains a W-W metallic bonding structure, it shows relatively better ductility compared to tungsten nitrides with a mixture of metallic and covalent bonding structures.
- (c) Experimental and theoretical evidence is listed below: (i) Our previous investigations also demonstrated that a body-centered cubic (BCC) structure could form when the deposited Ar:N₂ ratio is excessively low[2]. These BCC structures can be attributed to

metallic tungsten. Considering that the lattice constant of metallic BCC W ($a=3.16 \text{ \AA}$) is much smaller than that of FCC WN_x (approximately $4.3\text{--}4.4 \text{ \AA}$, depending on nitrogen-vacancy contents). Our SAED results (**Supplementary Figs S1**) confirm that the current coating does not contain any BCC-W phase. (ii) Our HRTEM (phase contrast image, **Fig.R1a**) and HAADF (z-contrast image, **Fig.R1b**) directly observe the W atom column positions in the $[110]$ projection. Here, the lattice angle between the two $\{111\}$ planes is close to 70.52 degrees, and the interplanar spacing is $\sim 2.49 \text{ \AA}$, which differs from those angles and spacings in the BCC-W structure. Therefore, these corroborate the rock-salt structure in the current WN_x . (iii) In addition, theoretical simulations [2-9] also rule out the presence of BCC structures in WN. The reported WN phases can include stoichiometric hexagonal $\delta\text{-WN}$, cubic NbO-type phase WN_x (W sits on FCC lattice), rock-salt $\text{WN}_{0.5}$, hexagonal and rhombohedral W_2N_3 .

To elaborate on phase structure, we add a section about electron diffraction characterization and new XRD results in the supplementary material (**Supplementary Fig.S1a**).

Fig.R1, HRTEM and HAADF observation WN_x in the $[110]$ projection.

(2) The bending strain is different from the normal strain and strongly depends on the geometry of the specimen. But, it reads like the authors implied a correlation of bending strain to toughness. As not a notch was made on the specimens, even a diamond nanoneedle could be superelastic (Science 360 (2018) 300). Even just in comparison of different samples, the high elastic bending strain doesn't correspond to high fracture toughness (particular the KIC). Since the data of fracture toughness were published in previous studies, the statements are suggested to be more focused on the deformation mechanism, i.e. the "vacancy-mediated" toughness, than the "toughness". Moreover, did the bent cantilever elastically recover, or with any residual plastic deformation? How would the authors comment on it from the perspective of elastic or plastic deformation?

We agree with the reviewer's point of view. For the cantilever experiment, high elastic deformation capacity (bending strain) indeed does not mean high fracture toughness (KIC).

However, the elevated elastic deformation capacity of TiN/WN_x leads to higher fracture energy compared with TiN. As the previous explanation [2], the normalized curves suggest a two times larger elastic deformability for the toughest TiN/WN_x superlattices compared to monolithic TiN. In combination with the increased critical stress intensity, this doubled elastic deformation limit contributes towards the substantially enhanced fracture energy (emulated by the shaded areas under the load-deflection curves).

We consider the current cantilever bends (without pre-notch) completely elastic deformations. We do not see residual plastic deformation for cantilever bending, a fact substantiated by comparing SEM images taken before and after elastic loading (**Fig.R2**). So far, small amounts of plastic deformation have solely been detected in the bending of cantilever beams in TMN/metal multilayer systems[10].

Although the postmortem observations under nanoindentation were under plastic deformation, the unit-cell disturbance mechanism we proposed is equally applicable to elastic deformation. Both AIMD and DFT simulations confirm that such disturbances emerge from the earliest stages of deformation and are not confined solely to the plastic phase. Characterizing the atomic-scale elastic response in real time is currently methodologically challenging. Nevertheless, we can infer the elastic response of WN_{0.5} by analyzing the elastic stress field around the dislocation core. HRTEM results show that WN_{0.5} has extremely severe unit cell disturbances near the dislocation core (enlarged HRTEM **Fig.6h**), while TiN shows a uniform shear-deformed lattice. These observations imply that unit cell disturbances can effectively mitigate the elastic stress field (Fig.6d and Supplementary Figs. S21), significantly mitigating the stress concentration effect during deformation.

Since the stress field affects the crack propagation, the presence of WN_x as a low-stress zone could direct the crack to deviate from high-stress areas, i.e., crack deflection (as seen in **Fig.R4** and **Supplementary Figs. S20**). Under elastic deformation to fracture, unit cell disturbances will dissipate the elastic stress/strain field, which increases the fracture energy to improve brittleness. Based on your comment, we have revised the discussion part (Pages 17-18) and added the supplementary material (**Supplementary Figs S20 and S21**).

Fig.R2, SEM view of the cantilever beam bending testing of WN_x/TiN superlattices. (c) shows the unloading image, where no residual plastic deformation is observed.

(3) A high local plastic strain of about 22% was observed in the indentations with a very low density of dislocations, but the strain might include not only plastic deformation but also the microcracking of the film or the delamination (or shear) at the interface, as typically observed in laminated ceramic coatings also with a low density of dislocations. How would the authors expect or observe? Has any similar evidence been seen in the bending tests? Or, would the authors consider to verify the high plastic strain by using micropillar compression?

We appreciate the reviewer's valuable comments!

The author's previous research unveiled that the indentation deformation of traditional TiN/AlN superlattices (SLs) indeed resulted in an exceedingly high dislocation density[11]. As the reviewer suggested scenario, plastic deformation within the TiN/AlN SL was concomitant with microcracking and shear slipping[12]. However, the deformation behavior of $TiN/WN_{0.5}$ discussed in this work starkly contrasts the cases mentioned above. We hardly detect the presence of microcracks, delamination, or shear at the interface, as shown in our HAADF image (**Fig.R3**).

Our SEM images of the fracture surface after the bending test also did not exhibit significant microcracking. However, distinct cleavage steps were discernible. We suggest that the more resilient $WN_{0.5}$ layer contributes to deflecting the crack during fracture (**Fig.R4**).

Using micropillar compression to verify that high plastic strain is indeed a feasible solution. However, the current coating thickness is too small to meet the prerequisites of micropillar compression experiments. Typically, the ratio of the diameter to height of micropillars necessitates a proportion of about 1:10. The micropillar compression experiments conducted by Kagerer et al. (under review, <http://dx.doi.org/10.2139/ssrn.4526735>) utilized a multilayer structure with a total thickness of approximately 6.0 μm , whereas the thickness in current work is only approximately 1.4 μm .

It's important to mention that an excessively diminutive micropillar diameter might lead to significant sample size effects, thereby influencing the ductile-brittle transition of the material. While the micropillar compression, cyclic loading, and TEM observations could indeed enhance the comprehensiveness of our investigation, the current study predominantly centers on pioneering research and analysis of novel deformation mechanisms. Future work holds the potential for expanding our study to encompass the mentioned micropillar experiments.

Fig.R3, A postmortem HAADF observation after nano-indentation with a load of 500 mN.

Fig.R4, SEM micrographs of fractured cross-sections of cantilever beams.

(4) Most importantly, the key point of this work is the enhanced toughness and improved plasticity through the function of the vacancy-caused lattice disturbances at a low dislocation density. It might be possible, and all the discussion is good (though the latter part is a little bit too long). But, these disturbances look immobile. The strain rate of typical crystalline materials can be estimated by the Orowan equation. If it is feasible that the atomic-level vacancy-caused unit-cell disturbances are the carriers of large-scale plasticity, could the authors estimate the density and mobility of these defects? Since atom arrangements are necessary for plastic deformation, for a very slight increase in the density of dislocations,

what amount and what scale of the short-range displacements would contribute towards the total large plastic strain?

Regarding the density of defects, our transmission electron microscopy (TEM) observations reveal a highly concentrated distribution of nitrogen vacancies in WN_x . Both electron energy loss spectroscopy (EELS) and energy-dispersive X-ray spectroscopy (EDAX) analysis indicate the absence of vacancy cluster structures within the WN_x layer. This observation suggests that vacancies are present in nearly every unit cell. Given homogenous distribution, we could estimate the vacancy density using EDXS /EELS, as shown in the main text. We do not consider our plastic deformation to require defect (vacancy) motion. AIMD simulations also did not show that the migration of vacancies to adjacent unit cells occurred during tensile deformation. Our proposed disturbances carried plasticity come from the volume change of the unit cell, which alters the mass density of the $WN_{0.5}$ layer. For plastic deformation, the total strain comes from the coupling of different volume cells (as shown by our DFT simulations).

Through our TEM observation and statistical analysis in indented thin film, the increased dislocation density is about 15% of the theoretical dislocation density. This contributes approximately less than 1% to the bending strain. In Fig. 4, we highlight that there are almost no dislocations in the HRTEM with a misorientation angle of less than 4 degrees, which can be understood as the disturbances of the unit cell contributing to most of the plastic strain. We note from images of the indented sample (Fig. S9) that the SL bending occurs via continuous and gradual lattice rotation at a slight angle (about $1.3^\circ/100$ nm, Fig. S9). This slight angle bending deformation suggests that the unit cell disturbances are the principal contributors to plasticity.

(5) Additionally, Figure 1c about the EELS spectra should be 1b. And, some characters in the figures are too small to be clearly seen.

Thank the reviewer for pointing out these issues. Now, we have made the corrections.

Reviewer #2 (Remarks to the Author):

In the manuscript NCOMMS-23-29233 authored by Chen et al., the atomistic microstructure of vacancy-mediated superlattice (SL) WN_x/TiN thin films is investigated, with a focus on its correlation with the mechanical properties. The study showcases cutting-edge experimental work, complemented by the implementation of ab initio molecular dynamics (AIMD) calculations. The work main contribution is the proposal of a novel deformation mechanism called "unit-cell disturbances." This mechanism is expected to show potential to enhance the elasticity, flexural strength, and fracture toughness of non-stoichiometric nanoceramics. This is of utmost

importance, as it presents a new pathway to tackle the longstanding strength and toughness tradeoff often observed in brittle materials. The analytical transmission electron microscopy (TEM) work conducted in this study deserves special praise. Particularly remarkable is the observation of vacancies along [100] and [110] directions, revealing an exquisite microstructural insight.

I find the main manuscript idea very progressive, particularly the concept of plastic deformation being mediated by unit-cell disturbances, as depicted in the schematic illustration of Fig. 4c. The authors propose that critical stress concentrations can be alleviated through localized unit cell deformations, leading to improved deformability and toughness at the macro-scale. The notion of creating a cellular material with atomistic voids, where the bonding character changes to a metallic one, is quite captivating. This innovative concept not only offers promising insights into studying the mechanical properties of non-stoichiometric "atomistic composites" but also opens up avenues to explore their other physical properties. I am particularly interested in the potential of this approach to study materials that lie at the threshold between dislocation- and unit-cell disturbance-mediated plasticity.

Regarding the Editor's inquiries, it is to note that the research group has previously published numerous works investigating the correlation between mechanical properties and microstructure of nano-laminates. Notably, Buchinger et al.'s paper [19] also demonstrated similar effects, showcasing enhanced mechanical properties through the use of superlattice (SL) TiN/WN architectures. The key differentiating factor in NCOMMS-23-29233, compared to the previous works by other authors, lies in the introduction of the concept of unit-cell disturbances. This approach sets the current manuscript apart, offering mainly a perspective to interpret the mechanical properties of nano-laminates with a large concentration of vacancies.

The observed hardness of ~ 37 GPa and fracture toughness of ~ 4.7 MPa $m^{0.5}$ reported in NCOMMS-23-29233 is comparable with the values reported already in the previous works of the authors, like 36.7 GPa and 4.6 MPa $m^{0.5}$ in Ref. [19], respectively, as well as by others in similar materials.

The concept of improved mechanical properties (by using element deficiency and excess) within non-stoichiometric nano-laminates is similar to that reported for self-assembled coherent cubic AlN/TiN [<https://doi.org/10.1016/j.actamat.2017.04.009>] for which also similar hardness and toughness values of 36 GPa and 4.7 MPa $m^{0.5}$, respectively, were observed [<https://doi.org/10.1039/C8NR10339A>].

So, the manuscript's primary contribution lies in the introduction of a new deformation concept, rather than presenting a new material type with a unique microstructure or mechanical properties.

The authors appreciate the reviewer's positive comments very much.

I have three concerns related to the manuscript main statement of unit cell disturbance-mediated deformation:

A) The authors state that "by introducing N vacancies, the bonding character changes to more metallic, i.e., is dominated by delocalized electrons that control the ability to redistribute stresses via plastic deformation". The plastic deformation means for me an irreversible change of the sample shape during deformation. When performing bending tests on the micro-cantilevers, did the authors observe irreversible change in the cantilever shape

(or the cantilever shape fully recovered after unloading)? The load-deflection curves in Fig. 3b indicate linear-elastic response without any "macroscopic" plasticity. The pile-ups in Fig. 3c could have also other origins than plastic flow.

Indeed, as illustrated by the load-deflection curves in Figure 3b, a linear-elastic response is evident, revealing the absence of "macroscopic" plasticity. For cantilever bending (without pre-notched), we did not observe any residual plastic deformation, which can be confirmed by comparing two SEM images before and after elastic loading (**Fig. R2**). Although the postmortem observations under nanoindentation were under plastic deformation, the unit-cell disturbance mechanism we proposed is equally applicable to elastic deformation. The simulation results confirm that such a disturbance occurs almost from the earliest stage of deformation and is not limited to the plastic stage. More crucially, these observations imply that unit cell disturbances can effectively mitigate the elastic stress field, significantly mitigating the stress concentration effect during deformation.

As for pile-up, previous studies suggested that such a pile-up produced by the indentation deformation of the cubic indenter can be considered a sign of its increased toughness and the existence of plastic flow. This can be referred to the research work given by Buchinger et al.[13] and Kindlund et al [14].

B) While the authors introduce a novel deformation mechanism to explain the improved elastic deformability observed in Fig. 3a,c, there remains however uncertainty about the fracture behavior of W_Nx/TiN , particularly the mechanism behind the linear-elastic fracture depicted in Fig. 3b. The nature of the fracture, whether it is transgranular or intercrystalline, is not entirely clear, especially considering that typically, TMN thin films tend to suffer from brittle fracture along grain boundaries or 2D-3D defects. Therefore, it raises questions about whether the W_Nx/TiN nanolaminate used in this study is indeed monocrystalline, free from 2D-3D defects. It would be beneficial if the revised manuscript includes a discussion on the expected fracture mechanism and a thorough analysis of the fracture surfaces of the cantilevers. It might be helpful to explore the presence of cleavage planes to gain further insights. Furthermore, addressing the preparation process of the monocrystalline W_Nx/TiN on MgO and discussing potential 2D and 3D structural defects (using conventional TEM?) would contribute to a more comprehensive understanding of the material's mechanical behavior. By providing additional clarity on the fracture mechanism and analyzing fracture surfaces, the authors could enhance the significance and reliability of their findings.

We appreciate the reviewer very much for these insightful suggestions.

In the revised manuscript (**Supplementary Figs S20**), we have added the SEM micrographs from fracture surfaces of un-notched free-standing samples. Following your comment, we have also discussed this part in the revised manuscript (Pages 17-18) and the supplementary material.

For the cantilever beam bending experiments, the observed fracture behavior showcases a significant toughness enhancement. As it is known, a macroscopic increase in toughness is linked to the augmented roughness of the fracture surface and, subsequently, to the overall fracture surface area[15]. Therefore, dissipating fracture energy through frequent crack deflection proves to be a practical approach to enhance toughness.

Crack deflection is experimentally demonstrated and shows a stepped fracture in **Fig. R5**. Since the fracture surface is very rough and the crack extends in a tortuous way. This indicates that WN_x can significantly deflect the crack. This can be explained by the unit cell disturbances mechanism proposed in this work. HRTEM observations and GPA analysis (**Fig.6d** and **Supplementary Figs S21**) show that unit cell disturbances can effectively attenuate the elastic stress field, significantly avoiding the stress concentration effect during deformation. The crack propagation path may be affected by the stress field. The presence of WN_x as a low-stress zone could direct the crack away from high-stress areas, thereby resulting in crack deflection.

For the preparation process (as shown in Supplementary Materials and Methods), we synthesized TiN/WN_x thin films with 10 nm periods (~ 5 nm/ ~ 5 nm) by unbalanced DC reactive magnetron sputtering using an AJA International Orion 5 magnetron sputtering deposition system. We used a 3"Ti target, and a 2" W target. The reactive magnetron sputtering process was carried out at 500°C (substrate temperature) in an Ar/N₂ mixed gas atmosphere with a total pressure of 0.4 Pa and an Ar/N₂ flow ratio of 5.3 sccm / 4.7 sccm. To avoid the intermixing of the two-layer materials via excessive ion bombardment, we applied a relatively low bias potential of -40 V to the MgO (100) substrate, just enough to obtain a dense coating morphology.

Addressing the matter of whether the superlattice (SL) constitutes a single crystal and the potential presence of 2D-3D defects, we have incorporated supplementary TEM and X-ray diffraction (XRD) experiments (as indicated in the revised **Supplementary Fig.S1** and below). The XRD patterns of TiN/WN_x coatings grown on MgO (100) platelets (**Fig. R6a**), depict a clear monocrystalline (100) texture. SEM cross-sectional and TEM-BF/DF observations also clearly show that our SL has no grain boundary structure (2D defect) or pores (3D defect). In addition, our large-scale SAED (aperture diameter is about 1 μ m) results also demonstrate the single-crystal character of the current TiN/WN_x film.

Fig.R5, SEM micrographs of fractured cross-sections of cantilever beams (un-notched sample). (a) and (c) are top view images, and (b) and (d) are the corresponding side view images, respectively.

Fig.R6. (a) X-ray diffraction patterns of SL coating deposited on MgO (100) (b) SEM micrographs of SL cross-sections. (c) Overall TEM-BF/DF observation of as-deposited SL. (d) SAED result of as-deposited SL. The aperture diameter is about 1.0 μm , which contains most of the film (film total thickness is about 1.4 μm).

C) It would be also interesting to analyze mechanical properties of nonstoichiometric WN_x , as a reference material, and compare them with the performance of the WN_x/TiN nanolaminate. Currently, it is not clear how the synergetic effect of the SL and the vacancies actually controls and influences the fracture. It would be genius to compare the present material with WN_x of high concentration of N vacancies to see the difference in the mechanical performance.

We concur with the viewpoint presented by the reviewer. However, achieving the synthesis of single-crystalline monolithic *rs*- WN_x with 50% N vacancies proves formidable. Tungsten nitride crystallizes in a wide range of phases with various anion-to-cation ratios, including cubic WN , W_2N , and W_3N_2 , hexagonal W_2N , W_5N_4 , and W_5N_8 , and rhombohedral W_7N_6 , which could form depending on the chosen processing parameters such as temperature and N_2 flow rate during deposition from the vapor phase [2-9]

Monolithic WN has been synthesized in several research groups. Ozsdolay et al. [16] showed the synthesized NbO-type phase WN (space group Pm-3m, with 25% of vacancies on both sublattices). Buchinger et al. tried to synthesize $rs\text{-WN}_x$ on a Si substrate by changing the ratio of Ar: N₂. However, their XRD results showed that the coating synthesized under lower N partial pressure contained metallic tungsten. Monolithic coatings synthesized at high N partial pressures cannot fully approach the theoretical lattice constant of $rs\text{-WN}_{0.5}$. Thus, the authors speculate that the monolithic WN_x may contain other non-stoichiometric phases, e.g., NbO-type WN phase. The introduction of other non-stoichiometric ratio phases has a very high impact on the mechanical properties of the coating. For example, DFT simulations[17] have shown that the hardness of the $\beta\text{-Mo}_2\text{N}$ (tetragonal phase) is almost the lowest among all the phases of MoN. Its theoretical hardness is only about 60% of the fcc γ' phase (rock-salt phase with 50% disordered N Vacancies). Given these considerations, we have focused on studying the TiN/WN_{0.5} superlattice system. The preference arises from the lower formation energy of the rock-salt phase with 50% disordered N vacancies within the superlattice structure. Simultaneously, rock-salt phase phonon vibrations are more stable than other phase configurations[2].

Indeed, the interface effect of SL is also an essential mechanism for improving the hardness and toughness of hard coatings. However, we do not believe the interface effect is the main contributor to enhancing the toughness of the current TiN/WN_{0.5} SL. Coherent interfaces give certain residual stresses, both tensile and compressive, surrounding the interfaces between two different layers. These stresses have to be overcome during cleavage. Thus, they may effectively enhance the measured fracture toughness. The residual stress/strain, i.e., misfit strain, at the coherent interface strongly depends on the different lattice constants of the two layers. For TiN $a=4.25\text{\AA}$ and WN_{0.5} $a=4.18\text{\AA}$ [2], the theoretical strain is only about 1.7%. This is much lower than conventional SL, e.g., TiN/AlN is about 3.2%. Experimental evidence can also be found in the GPA **Fig. R7** analysis (strain distribution in the out-of-plane direction) we provide below. Our GPA results did not reveal significant strain differences between the two layers. This indicates that the unit cell disturbances of the WN_{0.5} layer dissipate its interfacial coherent elastic stress/strain fields. Therefore, unit cell disturbances significantly reduce the impact of interfacial effects in TiN/WN_{0.5}.

Fig.R7, (a) HRTEM observation (viewed along the [001] direction) of the coherent interface of as-deposited TiN/AIN. (b) GPA analysis of the corresponding image in (a). (c) HRTEM observation (viewed along the [001] direction) of the coherent interface in as-deposited TiN/WN_x. (d) GPA analysis of the corresponding image in (c). The red-marked box in (b) and (d) are used for reference.

NC Inquiries:

- The correlation of micro-mechanical studies, analytical TEM and AIMD calculations is really nice and represents state of the art.
- The novelty of the work resides exclusively in introducing the concept of vacancies-mediated unit cell disturbance, which is stated to be responsible for better deformability of the nanolaminate. It can be expected that the work will attract the attention of the TMN community and stimulate new research. It appears that the work does not introduce a new material concept – non-stoichiometric superlattices and/or multilayered TMN microstructures with practically equal mechanical properties were already reported.
- Some statements, especially those related to the observed plastic deformation and linear-elastic fracture seems to be not justified. The authors could provide experimental evidence that the proposed unit cell disturbance mechanism improves also the fracture behavior of the nanolaminates – see the discussion (A-C) above.
- The used methodology is sound and the group work represents the state of the art in the field.
- The method description is sufficient and can be reproduced.

We appreciate it very much for the comments.

In this work, our intensive TEM analysis is performed based on postmortem observations of indented samples, and the proposed novel deformation mechanism can be linked to its plastic deformation. However, we believe the unit cell disturbances mechanism still applies to its elastic deformation. Here, we present four direct or indirect evidence to support this argument:

Evidence (1): Both AIMD and DFT simulations confirm that such disturbance occurs almost from the earliest stages of deformation and is not limited to the plastic stage.

Evidence (2): The cantilever beam bending experiment (elastic bending) shows the variation in elastic modulus, which exhibits a decay with strain and is accompanied by a slight fluctuation. The attenuation of about 3.5% modulus can be considered the equivalent elastic strains applied to WN_x trigger strongly inhomogeneous unit cell distortions. Local inhomogeneous response under load conditions leads to macroscopic-scale nonlinear deformation.

Evidence (3): Here, we can infer the elastic response of WN_x by analyzing the elastic stress field around the dislocation core. HRTEM results show that WN_x has extremely severe unit cell disturbances near the dislocation core (**Fig.6h**), while TiN shows a uniform shear-deformed lattice. This suggests that the elastic stress field near the dislocation core in $WN_{0.5}$ triggers this unit cell perturbation.

Evidence (4): In **Fig. R7**, we perform further strain analysis on the as-deposited $WN_{0.5}/TiN$ SL. Theoretically, the lattice misfit strain between the TiN ($a = 4.25\text{\AA}$)/ $WN_{0.5}$ ($a = 4.18\text{\AA}$) is about 1.7%, and $WN_{0.5}$ should have a compressive strain field in the out-of-plane direction. However, our GPA results did not reveal significant strain differences between the two layers. This indicates that the unit cell disturbances of the $WN_{0.5}$ layer dissipate its interfacial coherent elastic stress/strain field. We add the above evidence to the supplementary material (**Supplementary Figs S21**).

The mechanism involving disturbances remains relevant to linear-elastic fracture. In the revised manuscript, we have incorporated SEM micrographs of fracture surfaces from un-notched free-standing samples into it (as showcased in **Supplementary Figures S20**). This experimental evidence further suggests that the proposed unit cell disturbance mechanism improves fracture behavior. Crack deflection is experimentally demonstrated and exhibits a stepped fracture morphology. Since the fracture here is very rough and the crack extends in a tortuous way, we show that the introduction of WN can significantly deflect the crack. This can be explained by the unit cell disturbance effect on dissipating the elastic stress/strain field, which would lead to

the formation of low-stress regions within the WN_x layer upon elastic deformation. Since the crack propagation path may be affected by the stress field, and the presence of WN_x as a low-stress zone could direct the crack to deviate from high-stress areas, thereby resulting in crack deflection. This macroscopically increases the toughness by increasing the crack propagation distance.

Reviewer #3 (Remarks to the Author):

The authors elaborated the vacancies-mediated mechanical property enhancement for WN_x/TiN superlattices. The structural, compositional and nanomechanical analysis of WN_x/TiN SL architecture is clear and convincing. Therefore, I recommend the paper to be published after the following comments are taken into consideration:

The authors appreciate the reviewer's positive comments very much.

1. Fig. 1 (b) is mis-labelled as (c) for the EELS results.

Now, such issues have been corrected in the revised manuscript. See the Revised Figs. 1.

2. In row #192-193, it is mentioned that the actual dislocation density after deformation increases by $0.31 \times 10^{13}/\text{cm}^2$ which is much smaller than GND increase of $2.03 \times 10^{13}/\text{cm}^2$. But in the supplementary material Fig. S9, it is written that GND will increase $0.203 \times 10^{12}/\text{cm}^2$, which is much smaller than actual measurement. Which one is correct? If the supplementary material is correct, then the plasticity can be achieved solely by dislocation generation/motion.

Thank the reviewer for pointing out this issue. It is theoretically required that the GND density should increase by $2.03 \times 10^{13}/\text{cm}^2$. Now, this typo has been corrected in the revised supplementary material.

3. Could you explain why the unit-cell distortion will hinder the dislocation nucleation? If unit-cell distortion energy is much higher than dislocation formation energy, will dislocation forms instead of unit cell distortion?

In this work, we suggest that severe unit cell distortion and unit cell disturbances impede the nucleation of dislocations. (i) Experimentally, our atomic-scale observations and statistical analysis of the bending deformation in WN_x revealed that the dislocation density hardly agrees with the theoretical prediction from the Frank-Bilby equation (Figs.4). This important observation suggests that plastic deformation inhibits dislocation nucleation. (ii) From a

simulation perspective, our AIMD simulations observed dislocation nucleation events during the deformation of TiN, but such events were absent in WN_x.

Usually the emergence of dislocations typically hinges on the presence of well-defined slip systems within crystals. For example, conventional binary TMNs, e.g., TiN, coating has been studied in the past either by TEM or by calculating the crystallographic anisotropy using Schmid's law [18]. The primary slip system for dislocation glide in TiN crystals has been identified. Slip primarily occurs on {110} planes along the <110> directions [12, 18-20].

The suppression of dislocation nucleation due to unit cell distortion stems from the alteration it imposes on the atomic arrangement in the short-range order. This likely contributes to the absence of a fixed slip system, explaining why we did not observe the slip deformation mode in both experiments and simulations. Thermodynamically, DFT simulations of TMN materials (unpublished work by D. Holec et al.) found that introducing a large number of N vacancies (simultaneously accompanied by lattice distortion) increases the SFE. This actually supports the argument of the inhibition of dislocation nucleation. Salamania et al. [21] also show that vacancies inhibit dislocation nucleation and/or increase Peierls stresses via dislocation-pinning. To well address the questions posed by the reviewers, we also cite such previous DFT simulation results, see Page 19 for details.

We believe that dislocations will form if the unit cell distortion energy is higher than the dislocation formation energy. However, we think that the unit cell distortion energy in the WN_x system is much lower than the dislocation formation energy. This is confirmed by our DFT and AIMD simulations, where the unit cell distortion starts already in the early stage of elastic deformation, implying that the energy required for the unit cell is lower than the dislocation formation energy.

Reference

- [1] G. Zou, Y. Hong, S. Wang, S. Yin, S. Lei, Y. Wang, H. Zhu, T. Kuang, K. Zhou, Hard and tough nitrogen doped tungsten coatings deposited by HIPAC: Microstructure and mechanical properties, *Journal of Alloys and Compounds* 876 (2021) 160146.
- [2] J. Buchinger, N. Koutná, Z. Chen, Z. Zhang, P.H. Mayrhofer, D. Holec, M. Bartosik, Toughness enhancement in TiN/WN superlattice thin films, *Acta Materialia* 172 (2019) 18-29.
- [3] B.D. Ozsdolay, C.P. Mulligan, M. Guerette, L. Huang, D. Gall, Epitaxial growth and properties of cubic WN on MgO(001), MgO(111), and Al₂O₃(0001), *Thin Solid Films* 590 (2015) 276-283.
- [4] J.W. Klaus, S.J. Ferro, S.M. George, Atomic Layer Deposition of Tungsten Nitride Films Using Sequential Surface Reactions, *Journal of The Electrochemical Society* 147(3) (2000) 1175.
- [5] T. Polcar, N.M.G. Parreira, A. Cavaleiro, Structural and tribological characterization of tungsten nitride coatings at elevated temperature, *Wear* 265(3) (2008) 319-326.

- [6] M. Bereznoi, Z. Tóth, A.P. Caricato, M. Fernández, A. Luches, G. Majni, P. Mengucci, P.M. Nagy, A. Juhász, L. Nánai, Reactive pulsed laser deposition of thin molybdenum- and tungsten-nitride films, *Thin Solid Films* 473(1) (2005) 16-23.
- [7] M.L. Addonizio, A. Castaldo, A. Antonaia, E. Gambale, L. Lemmo, Influence of process parameters on properties of reactively sputtered tungsten nitride thin films, *Journal of Vacuum Science & Technology A* 30(3) (2012).
- [8] K. Balasubramanian, S. Khare, D. Gall, Vacancy-induced mechanical stabilization of cubic tungsten nitride, *Physical Review B* 94(17) (2016) 174111.
- [9] Z. Zhao, K. Bao, D. Duan, F. Tian, Y. Huang, H. Yu, Y. Liu, B. Liu, T. Cui, The low coordination number of nitrogen in hard tungsten nitrides: a first-principles study, *Physical Chemistry Chemical Physics* 17(20) (2015) 13397-13402.
- [10] A.K. Mishra, H. Gopalan, M. Hans, C. Kirchlechner, J.M. Schneider, G. Dehm, B.N. Jaya, Strategies for damage tolerance enhancement in metal/ceramic thin films: Lessons learned from Ti/TiN, *Acta Materialia* 228 (2022) 117777.
- [11] Z. Chen, Y. Zheng, L. Löfler, M. Bartosik, G.K. Nayak, O. Renk, D. Holec, P.H. Mayrhofer, Z. Zhang, Atomic insights on intermixing of nanoscale nitride multilayer triggered by nanoindentation, *Acta Materialia* 214 (2021) 117004.
- [12] Z. Chen, Y. Zheng, Y. Huang, Z. Gao, H. Sheng, M. Bartosik, P.H. Mayrhofer, Z. Zhang, Atomic-scale understanding of the structural evolution of TiN/AlN superlattice during nanoindentation—Part 1: Deformation, *Acta Materialia* 234 (2022) 118008.
- [13] J. Buchinger, N. Koutná, A. Kirnbauer, D. Holec, P.H. Mayrhofer, Heavy-element-alloying for toughness enhancement of hard nitrides on the example Ti-W-N, *Acta Materialia* 231 (2022) 117897.
- [14] H. Kindlund, D.G. Sangiovanni, J. Lu, J. Jensen, V. Chirita, J. Birch, I. Petrov, J.E. Greene, L. Hultman, Vacancy-induced toughening in hard single-crystal V_{0.5}Mo_{0.5}N_x/MgO(001) thin films, *Acta Materialia* 77 (2014) 394-400.
- [15] A.G. Evans, Perspective on the Development of High-Toughness Ceramics, *Journal of the American Ceramic Society* 73(2) (1990) 187-206.
- [16] B.D. Ozsdolay, C.P. Mulligan, K. Balasubramanian, L. Huang, S.V. Khare, D. Gall, Cubic β -WN_x layers: Growth and properties vs N-to-W ratio, *Surface and Coatings Technology* 304 (2016) 98-107.
- [17] H.S. Abdelkader, A. Rabahi, M. Benaissa, M.K. Benabadi, Theoretical investigation of structural and mechanical stability of Mo₂N, *Solid State Communications* 314-315 (2020) 113919.
- [18] M. Odén, H. Ljungcrantz, L. Hultman, Characterization of the Induced Plastic Zone in a Single Crystal TiN(001) Film by Nanoindentation and Transmission Electron Microscopy, *Journal of Materials Research* 12(8) (2011) 2134-2142.
- [19] L. Hultman, M. Shinn, P.B. Mirkarimi, S.A. Barnett, Characterization of misfit dislocations in epitaxial (001)-oriented TiN, NbN, VN, and (Ti,Nb) N film heterostructures by transmission electron microscopy, *Journal of Crystal Growth* 135(1) (1994) 309-317.
- [20] N. Koutná, L. Löfler, D. Holec, Z. Chen, Z. Zhang, L. Hultman, P.H. Mayrhofer, D.G. Sangiovanni, Atomistic mechanisms underlying plasticity and crack growth in ceramics: a case study of AlN/TiN superlattices, *Acta Materialia* 229 (2022) 117809.
- [21] J. Salamaia, D.G. Sangiovanni, A. Kraych, K.M. Calamba Kwick, I.C. Schramm, L.J.S. Johnson, R. Boyd, B. Bakhit, T.W. Hsu, M. Mrovec, L. Rogström, F. Tasnádi, I.A. Abrikosov, M. Odén, Elucidating dislocation core structures in titanium nitride through high-resolution imaging and atomistic simulations, *Materials & Design* 224 (2022) 111327.

REVIEWER COMMENTS

Reviewer #1 (Remarks to the Author):

The comments raised by the reviewer been addressed.

Reviewer #2 (Remarks to the Author):

Thank you for your responses to my previous inquiries!

As I mentioned in my initial review, I appreciate the concept of "unit-cell disturbances" and its potential to enhance the mechanical properties of TMN thin films. The primary contribution of this manuscript lies in introducing this novel deformation concept, rather than presenting a new material type with unique microstructures or mechanical properties. The measured SL does not exhibit any distinctive mechanical properties. This appears OK, because it was not the main aim to introduce a new material.

The analytical transmission electron microscopy (TEM) work carried out in this study is truly impressive.

However, the first sentence of the abstract and the central message of the manuscript seem to be centered on improving toughness, a critical material property within the TMN community's focus. Nonetheless, this main argument that vacancy-mediated superlattice (SL) W_{N_x}/TiN thin films exhibit superior elasticity, flexural strength, and (fracture) toughness lacks comprehensive documentation in the manuscript, particularly regarding the (fracture) toughness enhancement, which remains speculative.

Well-established techniques, such as bending notched cantilevers, are commonly used by the TMN community to determine fracture toughness in brittle thin films. This approach have not been applied.

The only indirect experimental proof that the SL in manuscript show an interesting deformability and possibly also superior fracture toughness is the fracture toughness is Fig.S8. The deformability appears really unique, I have never seen similar results from TMN thin films.

In principle, the manuscript can be accepted in the present conditions, even though some statements are not fully justified. It is the editor responsibility to make decision.

In case the manuscript is revised, following major issues should be addressed:

The reported "unit-cell disturbances" concept, while intriguing, lacks thorough demonstration in the manuscript. Furthermore, the observed irreversible deformation of the SL after indentation (Fig.S8) does not necessarily guarantee enhanced toughness.

The reported fracture strength values appear similar to those previously reported by the authors and others in prior publications. However, when comparing with the literature, it is important to acknowledge that all cited reports pertain to polycrystalline ceramics, where fractures typically occur along brittle columnar grain boundaries. In contrast, the monocrystalline SL in the current work should theoretically possess superior mechanical properties compared to polycrystalline ceramics tested in previous studies.

The authors exclusively tested unnotched cantilevers, yet their discussion predominantly revolves around toughness and toughening mechanisms, which are inherently more applicable to notched specimens. The rationale for not testing notched cantilevers is not clearly explained. Employing notched specimens would allow for a straightforward quantification of fracture toughness enhancement and stored strain energy.

Since unnotched cantilevers were used, it would be valuable to estimate the material's modulus. According to Griffith's equation (although it is a simplified model), toughness scales with the square root of the Young's modulus. Could you make this estimate?

The SEM micrographs, particularly Fig. S1 and Fig.S8, indicate that the SL is not homogeneous and appears to consist of sub-layers with varying compositions/period, with density increasing toward the film surface. Could this inhomogeneity potentially explain the observed crack deflection phenomenon?

Minor comments:

- P8, r156-163 it is no novelty, that ceramics may perform plastically under compression loading.
- P3, 68-72. Isn't the key motivation for this study to enhance plasticity? Here it is stated that "dislocation nucleation/motion is hindered by the unit-cell distortions". There is no need for unit cell distortions to hinder dislocation motion in ceramics, it is per se highly unlikely.
- P2, 42 spelling error – pores instead of porous
- P3, 64 there were no "nanomechanical experiments" just microcantilever bending tests.

- P7-8. The authors present the analytical solutions for a Euler-Bernoulli cantilever peak stress and strain. Why is there a need for a FE model? It appears actually superficial, but it can stay in the paper. Poisson's ratio of 0.39 seems quite an exaggeration for a ceramic material ($\nu \sim 0.2$).

-On P8, r154-155 it is claimed that "WNx SL has an excellent capacity to absorb strain energy". The concept of the modulus of resilience is valid for uniaxial tension, where stress and strain are independent of the position in the sample. Thus, the volumetric strain energy stored in a cantilever until fracture may differentiate quite significantly from a uniaxial tension specimen.

Reviewer #3 (Remarks to the Author):

The authors have resolved the questions raised in the previous comments, therefore, I recommend the paper to be published as it is.

Reviewer #2 (Remarks to the Author):

Thank you for your responses to my previous inquiries!

As I mentioned in my initial review, I appreciate the concept of "unit-cell disturbances" and its potential to enhance the mechanical properties of TMN thin films. The primary contribution of this manuscript lies in introducing this novel deformation concept, rather than presenting a new material type with unique microstructures or mechanical properties. The measured SL does not exhibit any distinctive mechanical properties. This appears OK, because it was not the main aim to introduce a new material.

The analytical transmission electron microscopy (TEM) work carried out in this study is truly impressive.

The authors appreciate the reviewer's comments very much. However, we still believe the measured SL exhibits distinctive mechanical properties. The elasticity and flexural strength are significantly higher than those of stoichiometric ceramics.

However, the first sentence of the abstract and the central message of the manuscript seem to be centered on improving toughness, a critical material property within the TMN community's focus. Nonetheless, this main argument that vacancy-mediated superlattice (SL) $W\text{N}_x/\text{TiN}$ thin films exhibit superior elasticity, flexural strength, and (fracture) toughness lacks comprehensive documentation in the manuscript, particularly regarding the (fracture) toughness enhancement, which remains speculative.

Well-established techniques, such as bending notched cantilevers, are commonly used by the TMN community to determine fracture toughness in brittle thin films. This approach have not been applied.

The only indirect experimental proof that the SL in manuscript show an interesting deformability and possibly also superior fracture toughness is the fracture toughness is Fig.S8. The deformability appears really unique, I have never seen similar results from TMN thin films. In principle, the manuscript can be accepted in the present conditions, even though some statements are not fully justified. It is the editor responsibility to make decision.

We value the comments from the reviewer. The fracture toughness of the current sample has been reported in our previous work[1]. Consequently, this study does not encompass fracture toughness measurements with pre-notched samples. Given that the fracture toughness measured with pre-notched samples in our prior study reached $4.7 \text{ MPa} \sqrt{\text{m}}$, we believe that the claims of high fracture toughness in this work are not speculative. However, to fulfill the reviewer's comments, we tone down a bit about the mechanical property claims in the manuscript.

Recognizing that the primary innovation in this study lies in the proposal of a novel deformation mechanism and considering the feedback from the reviewer and editor, we have toned down a bit about the claim of improved toughness in the revised manuscript. We have made minor revisions in the abstract, introduction, and conclusion to emphasize our proposed novel deformation mechanism. In addition, we also weakened the comparison of mechanical properties (flexural strength elasticity and elastic strain energy) in the revised manuscript.

In case the manuscript is revised, following major issues should be addressed:

The reported "unit-cell disturbances" concept, while intriguing, lacks thorough demonstration in the manuscript.

In our experimental work, we substantiated the concept of unit-cell disturbances through HRTEM characterization at the atomic scale. Furthermore, our AIMD simulations provided additional confirmation of the TEM findings. Nevertheless, it should be noted that HRTEM analysis solely characterizes the distortion at a very localized level. We acknowledge the lack of a sufficient demonstration, as the reviewer pointed out.

We added the analysis of electron diffraction patterns to demonstrate it in the revised manuscript. Through extensive SAED characterization, we conducted electron diffraction patterns at multiple locations after nanoindentation deformation. The distortion in the crystal lattice (or even in the unit cell) can result in the broadening of diffraction spots. Based on this, we inversely evaluate the distortions by qualitatively analyzing the extent of broadening in the (111) diffraction spot before and after deformation. The broadening of the (111) diffraction spot in the deformed superlattice (SL) is more significant than that observed in the coating before deformation. Our HRTEM observations and statistical analysis did not reveal a significant increase in dislocation density. HRTEM interplanar spacing measurements also did not detect notable residual strain (not presented in this work). Therefore, the broadening here can eliminate the effects of dislocations and lattice strain. On a larger scale, we present evidence of deformation-induced unit cell disturbances. To be clear, we added a section about electron diffraction analysis in the supplementary material (Fig.S16).

Fig.R1, Comparison of electron diffraction broadening before and after deformation.

Furthermore, the observed irreversible deformation of the SL after indentation (Fig.S8) does not necessarily guarantee enhanced toughness.

We agree with the reviewer that "the observed irreversible deformation after indentation of SL (Figure S8) does not necessarily guarantee enhanced toughness". Fig. S8 only confirms the existence of local compression plasticity under indentation deformation and we do not use Fig. S8 as an evidence of

toughness improvement. Here only shows that WN_x/TiN SL has a certain plastic deformation ability under compressive load. In the revised manuscript, we further emphasize this point.

The reported fracture strength values appear similar to those previously reported by the authors and others in prior publications. However, when comparing with the literature, it is important to acknowledge that all cited reports pertain to polycrystalline ceramics, where fractures typically occur along brittle columnar grain boundaries. In contrast, the monocrystalline SL in the current work should theoretically possess superior mechanical properties compared to polycrystalline ceramics tested in previous studies.

The author agrees with the reviewer. In revised Table S1, we further specified the fracture toughness of monocrystalline TMN. In general, polycrystalline TMN coatings tend to exhibit lower fracture toughness. However, the current WN_x/TiN SL demonstrates higher fracture toughness than traditional monocrystalline TMN coatings.

As demonstrated in our previous study, the fracture toughness of the monocrystalline WN_x/TiN SL (5nm/5nm, as studied here) is approximately 60% higher than that of monocrystalline TiN. Additionally, DFT simulations highlight the significant impact of introducing N vacancies in monocrystalline TiN/MN* SL on intrinsic fracture toughness. For instance, the Cauchy pressure of TiN/ $WN_{0.5}$ can reach 160 GPa, while that of TiN/WN (NbO) is -20 GPa[1]. For more simulation evidence, please refer to the simulation results by Koutná et al.[2].

In sum, we believe synthetic monocrystalline coatings have advantages in maintaining high fracture toughness, but introducing N vacancies will further improve the fracture toughness.

The authors exclusively tested unnotched cantilevers, yet their discussion predominantly revolves around toughness and toughening mechanisms, which are inherently more applicable to notched specimens. The rationale for not testing notched cantilevers is not clearly explained. Employing notched specimens would allow for a straightforward quantification of fracture toughness enhancement and stored strain energy.

As we mentioned above, the fracture toughness of the current samples has been shown in our previous work. Therefore, we did not include fracture toughness measurements under pre-notch in this work in order to show new experimental data. To consider the suggestions of the reviewer and editor, we tone down the statements of improvements in fracture toughness.

Since unnotched cantilevers were used, it would be valuable to estimate the material's modulus. According to Griffith's equation (although it is a simplified model), toughness scales with the square root of the Young's modulus. Could you make this estimate?

We appreciate the reviewer's valuable comments.

The elastic modulus E is calculated from the slope of the load-deflection curves as follows,

$$E = \frac{4}{B} \times \frac{dp}{dw} \times \left(\frac{L}{t}\right)^3$$

Where P and w are the load and deflection of the cantilever during loading, and B , L , and t are the cantilever's width, length, and thickness, respectively. The elastic modulus calculated by the Euler-Bernoulli formula is 299.6 ± 12.7 GPa. Using the finite element simulation, the fitted elastic modulus is

302 GPa. We added the derived elastic modulus to the revised manuscript. This high elastic modulus under bending cantilever test is way higher than the most nitride coatings.

The SEM micrographs, particularly Fig. S1 and Fig.S8, indicate that the SL is not homogeneous and appears to consist of sub-layers with varying compositions/period, with density increasing toward the film surface. Could this inhomogeneity potentially explain the observed crack deflection phenomenon?

Thanks to the reviewer for pointing out the issue. In the case of the current SL sample, there is indeed some fluctuation in the thickness of the TiN layer across approximately a dozen layers. However, the periodic thickness remains relatively constant near the substrate region. The perfect constant period region is 550 nm thick (as seen in **Fig.R2**), accounting for approximately one-third of the total film thickness.

Upon examining the cross-section post-fracture, it becomes evident that the residual cleavage steps are positioned at approximately 250-270 nm from the substrate surface. This observation highlights that significant crack deflection occurs within the constant period region, unaffected by the fluctuations in the TiN layer thickness.

Fig.R2, TEM observation of as-deposited SL. Two SEM images show the micrograph of fractured cross-sections of cantilever beams.

Minor comments:

- P8, r156-163 it is no novelty, that ceramics may perform plastically under compression loading.

In the revised manuscript, we have modified and diluted the relevant statements.

- P3, 68-72. Isn't the key motivation for this study to enhance plasticity? Here it is stated that "dislocation nucleation/motion is hindered by the unit-cell distortions". There is no need for unit cell distortions to hinder dislocation motion in ceramics, it is per se highly unlikely.

We have revised the statement. Instead of "the effect of unit-cell dislocation to dislocation nucleation," we now describe it as "the hindrance of unit-cell distortion to dislocation nucleation." However, it's important to note that enhancing dislocation mobility doesn't necessarily equate to increased plasticity in SL ceramics. A substantial quantity of dislocations might not have the capability to migrate over long

distances. Consequently, intensifying dislocation motion could potentially result in the accumulation of dislocations, leading to stress concentration and, ultimately, fracture.

-P2, 42 spelling error – pores instead of porous

-P3, 64 there were no "nanomechanical experiments" just microcantilever bending tests.

Thank the reviewer for pointing out these issues. The above issues have been corrected.

- P7-8. The authors present the analytical solutions for a Euler-Bernoulli cantilever peak stress and strain. Why is there a need for a FE model? It appears actually superficial, but it can stay in the paper. Poisson's ratio of 0.39 seems quite an exaggeration for a ceramic material ($\nu \sim 0.2$).

We want to emphasize that the cantilever beam experiment involves not purely bending deformation. Under cantilever bending, the cross-section experiences not only normal stress but also shear stress, leading to a non-planar deformation of the cross-section of the sample. Additionally, there may be normal stress in the longitudinal direction due to cantilever bending. As a result, the Euler-Bernoulli formula, which disregards these effects, is an approximation. The results obtained using the Euler-Bernoulli formula may have a very slight deviation from the actual values (the fracture strength demonstrated by Euler-Bernoulli is 8.83 GPa, while the finite element simulation value is 9.08 GPa.). Finite element analysis considers these effects, giving more accurate results.

Poisson's ratio of 0.39 is adopted based on the DFT simulation results [1]. Theoretically predicted high Poisson's ratio (>0.3) of TMN materials also include $\text{Ti}_{1-x}\text{Ta}_x\text{N}$ (0.304)[4], TaN (0.335)[3], NbN (0.319)[3], ZrN(0.320)[5] and NbN (0.33)[6]. Experimentally measured Poisson's ratio of TMN (HfN, by XRD) can also reach 0.35 [7]. We thus believe that 0.39 is also a reasonable value.

-On P8, r154-155 it is claimed that "WNx SL has an excellent capacity to absorb strain energy". The concept of the modulus of resilience is valid for uniaxial tension, where stress and strain are independent of the position in the sample. Thus, the volumetric strain energy stored in a cantilever until fracture may differentiate quite significantly from a uniaxial tension specimen.

Thank the reviewer for pointing out our issue. In the revised manuscript, we deleted the relevant statements and comparisons.

Reference

[1] J. Buchinger, N. Koutná, Z. Chen, Z. Zhang, P.H. Mayrhofer, D. Holec, M. Bartosik, Toughness enhancement in TiN/WN superlattice thin films, *Acta Materialia* 172 (2019) 18-29.

[2] N. Koutná, A. Brenner, D. Holec, P.H. Mayrhofer, High-throughput first-principles search for ceramic superlattices with improved ductility and fracture resistance, *Acta Materialia* 206 (2021) 116615.

[3] C. Hu, J. Zhang, L. Chen, Y.X. Xu, Y. Kong, J.W. Du, P.H. Mayrhofer, Self-layering of (Ti,Al)N by interface-directed spinodal decomposition of (Ti,Al)N/TiN multilayers: First-principles and experimental investigations, *Materials & Design* 224 (2022) 111392.

[4] P. Ou, J. Wang, S. Shang, L. Chen, Y. Du, Z.-K. Liu, F. Zheng, A first-principles study of structure, elasticity and thermal decomposition of $\text{Ti}_{1-x}\text{TM}_x\text{N}$ alloys (TM=Y, Zr, Nb, Hf, and Ta), *Surface and Coatings Technology* 264 (2015) 41-48.

[5] K. Bouamama, P. Djemia, D. Faurie, G. Abadias, Structural and elastic properties of single-crystal and polycrystalline $\text{Ti}_{1-x}\text{Zr}_x\text{N}$ alloys: A computational study, *Journal of Alloys and Compounds* 536 (2012) S138-S142.

[6] P. Ren, K. Zhang, X. He, S. Du, X. Yang, T. An, M. Wen, W. Zheng, Toughness enhancement and tribochemistry of the Nb-Ag-N films actuated by solute Ag, *Acta Materialia* 137 (2017) 1-11.

[7] A.J. Perry, A contribution to the study of poisson's ratios and elastic constants of TiN, ZrN and HfN, Thin Solid Films 193-194 (1990) 463-471.

REVIEWERS' COMMENTS

Reviewer #2 (Remarks to the Author):

My comments were addressed and I recommend the publication.